# Neural Analysis and Synthesis: Reconstructing Speech from Self-Supervised Representations

**Hyeong-Seok Choi**[1,4]   **Juheon Lee**[1,4]   **Wansoo Kim**[1,4]

**Jie Hwan Lee**[4]   **Hoon Heo**[4]   **Kyogu Lee**[1,2,3,4]

[1]MARG, Department of Intelligence and Information, Seoul National University   [2]GSAI   [3]AIIS
[4]Supertone Inc.
{kekepa15, juheon2, wansookim, kglee}@snu.ac.kr, {wiswisbus, hoon}@supertone.ai

## Abstract

We present a neural analysis and synthesis (NANSY) framework that can manipulate voice, pitch, and speed of an arbitrary speech signal. Most of the previous works have focused on using information bottleneck to disentangle analysis features for controllable synthesis, which usually results in poor reconstruction quality. We address this issue by proposing a novel training strategy based on information perturbation. The idea is to perturb information in the original input signal (e.g., formant, pitch, and frequency response), thereby letting synthesis networks selectively take essential attributes to reconstruct the input signal. Because NANSY does not need any bottleneck structures, it enjoys both high reconstruction quality and controllability. Furthermore, NANSY does not require any labels associated with speech data such as text and speaker information, but rather uses a new set of analysis features, i.e., wav2vec feature and newly proposed pitch feature, Yingram, which allows for fully self-supervised training. Taking advantage of fully self-supervised training, NANSY can be easily extended to a multilingual setting by simply training it with a multilingual dataset. The experiments show that NANSY can achieve significant improvement in performance in several applications such as zero-shot voice conversion, pitch shift, and time-scale modification [1].

## 1   Introduction

Analyzing and synthesizing an arbitrary speech signal is inarguably a significant research topic that has been studied for decades. Traditionally, this has been studied in the digital signal processing (DSP) field using fundamental methods such as sinusoidal modeling or linear predictive coding (LPC), and it is the analysis and synthesis framework that lies at the heart of those fundamental methods [20, 3]. These traditional methods, however, are limited in terms of controllability because the decomposed representations are still low-level representations. It is obvious that the closer we decompose a signal into high-level/interpretable representations, the more we gain access to the controllability. Given this consideration, we aim to design a neural analysis and synthesis (NANSY) framework by decomposing a speech signal into analysis features that represent pronunciation, timbre, pitch, and energy. The decomposed representations can be manipulated and re-synthesized, enabling users to manipulate speech signals in various ways.

It is worth noting that many similar ideas have been recently proposed in the context of voice conversion applications. We categorize the previous works in two ways, i.e., 1. Text-based approach,

---

[1]audio samples: https://tinyurl.com/eytw7hmb

35th Conference on Neural Information Processing Systems (NeurIPS 2021).

2. Information bottleneck approach. The first approach exploits the fact that the text modality is inherently disentangled from the speaker identity. One of the most popular text-based approaches is to use a pre-trained automatic speech recognition (ASR) network to extract a phonetic posteriogram (PPG) and use it as a linguistic feature [41]. Then, combining the PPG with the target speaker information, the features are re-synthesized to a speech signal. Another alternative approach is to directly use text scripts by aligning it to a paired source signal [33]. Although these ideas have shown promising results, it is important to note that these approaches have common problems. First, in order to extract the PPG features, it is required to train an ASR network in a supervised manner, which demands a lot of paired text and waveform datasets. Additionally, the language dependency of the ASR network limits the model's capability to be extended to multilingual settings or languages with low-resources. To address these concerns, efforts have been made to divert from using the text information and the most popular approach is to use an information bottleneck. The key idea is to restrict the information flow by reducing time/channel dimension, and normalizing/quantizing intermediate representations [36, 10, 51]. Although these ideas have been explored in many ways, one critical problem is that there exists an inevitable trade-off between the degree of disentanglement and the reconstruction quality. In other words, there is a trade-off between speaker similarity and the preservation of original content such as linguistic and pitch information.

To avoid the major concern of the text-based approach, we suggest to use two analysis features which are wav2vec and a newly proposed feature, Yingram. In order to preserve the linguistic information without any text information, we utilize wav2vec 2.0 [4], trained on 53 languages in total [11]. While the features from wav2vec 2.0 have mostly been used for a downstream task, we seek the possibility of using them for an upstream/generation task. In addition, we propose a new feature that can effectively represent and control pitch information. Although it is the fundamental frequency ($f_0$) that is mostly used to represent the pitch information, $f_0$ is sometimes ill-defined when there exists sub-harmonics in the signal (e.g., vocal fry) [16, 1, 2]. We address this issue by proposing a controllable but more abstract feature than $f_0$ that still includes information such as sub-harmonics. Because the proposed feature is heavily inspired by the famous Yin algorithm [13], we refer to this feature as Yingram.

Although the analysis features above have enough information to reconstruct the original speech signal, we have found that the information in the proposed analysis features share common information such as pitch and timbre. To disentangle the common information so each feature can control a specific attribute for its desired purpose (e.g., wav2vec $\rightarrow$ linguistic information only, Yingram $\rightarrow$ pitch information only), we propose an information perturbation approach, a simple yet effective solution to this problem. The idea is to simply perturb all the information we do not want to control from the input features, thereby training the neural network to not extract the undesirable attributes from the features. Through this way, the model no longer suffers from the unavoidable trade-off between reconstruction quality and feature disentanglement, unlike the information bottleneck approach.

Lastly, we would like to deal with unseen languages at test time. To this end, we propose a new test-time self-adaptation (TSA) strategy. The proposed self-adaptation strategy does *not* fine-tune the model parameters but only the input linguistic feature, which consequently modifies the mispronounced parts of the reconstructed sample. Because the proposed TSA requires only a single sample at test-time, it adds a large flexibility for the model to be used in many scenarios (e.g., low-resource language).

The contributions of this paper are as follows:

- We propose a neural analysis and synthesis (NANSY) framework that can be trained in a fully self-supervised manner (no text, no speaker information needed). The proposed method is based on a new set of analysis features and information perturbation.

- The proposed model can be used for various applications, including zero-shot voice conversion, formant preserving pitch shift, and time-scale modification.

- We propose a new test-time self-adaptation (TSA) technique than can be used even on unseen languages using only a single test-time speech sample.

## 2 NANSY: Neural Analysis and Synthesis

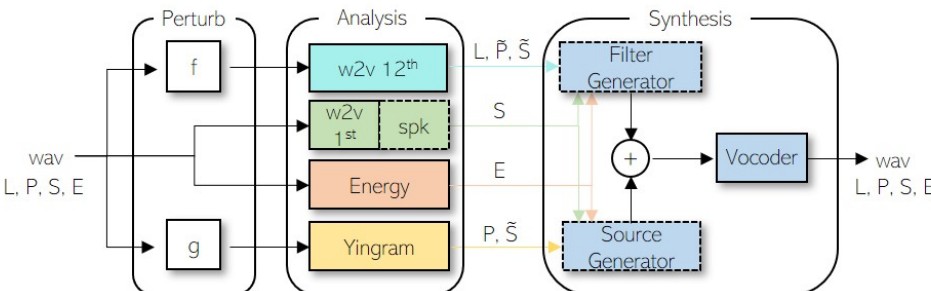

Figure 1: The overview of the training procedure and information flow of the proposed neural analysis and synthesis (NANSY) framework. The waveform is first perturbed using functions $f$ and $g$. $f$ perturbs formant, pitch, and frequency response. $g$ perturbs formant and frequency response while preserving pitch. w2v denotes wav2vec encoder and spk denotes a speaker embedding network. L, P, S, E denotes Linguistic, Pitch, Speaker, and Energy information, respectively. The tilde symbol is attached when the information is perturbed using the perturbation functions. The dashed boxes denote the modules that are being trained.

### 2.1 Analysis Features

**Linguistic** To reconstruct an intelligible speech signal, it is crucial to extract rich linguistic information from the speech signal. To this end, we resort to XLSR-53: a wav2vec 2.0 model pre-trained on 56k hours of speech in 53 languages [11]. The extracted features from XLSR-53 have shown superior performance on downstream tasks such as ASR, especially on low-resource languages. We conjecture, therefore, that the extracted features from this model can provide language-agnostic linguistic information. Now the question is, from which layer should the features be extracted? Recently, it has been reported that the representation from different layers of wav2vec 2.0 exhibit different characteristics. Especially, Shah et al. [39] showed that it is the output from the middle layer that has the most relevant characteristics to pronunciation[2]. In light of this empirical observation, we decided to use the intermediate features of XLSR-53. More specifically, we used the output from the 12th layer of the 24-layer transformer encoder.

**Speaker** Perhaps the most common approach to extract speaker embeddings is to first train a speaker recognition network in a supervised manner and then reuse the network for the generation task, assuming that the speaker embedding from the trained network can represent the characteristics of unseen speakers [21, 36]. Here we would like to take one step further and assume that we do not have speaker labels to train a speaker recognition network in a supervised manner. To mitigate this disadvantage, we again use the representation from XLSR-53, which makes the proposed method fully self-supervised. To determine which layer of XLSR-53 to extract the representation from, we first analyzed the features from each layer. Specifically, we averaged the representation of each layer along the time-axis and visualized utterances of 20 randomly selected speakers from the VCTK dataset using TSNE [47, 46]. In Fig. 2, we can observe that the representation from the 1st layer of XLSR-53 already forms clusters for each speaker, while the latter layers (especially the last layer) tend to lack them. Note that this is in accordance with the previous observation in [18]. Taking this into consideration, we train a speaker embedding network that uses the 1st layer of XLSR-53 as an input. For the speaker embedding network, we borrow the neural architecture from a state-of-the-art speaker recognition network [14], which is based on 1D-convolutional neural networks (1D-CNN) with an attentive statistics pooling layer. The speaker embedding was $L_2$-normalized before conditioning. The speaker embeddings of seen and unseen speakers during training are also shown in Fig. 2.

**Pitch** Due to the irregular periodicity of the glottal pulse, we often hear creaky voice in speech, which is usually manifested as jitter or sub-harmonics in signals. This makes hard for $f_0$ trackers to estimate $f_0$ because the $f_0$ itself is not well defined in such cases [16, 1, 2]. We take a hint from the

---

[2]We failed to train the model when we used the wav2vec features from the last three layers.

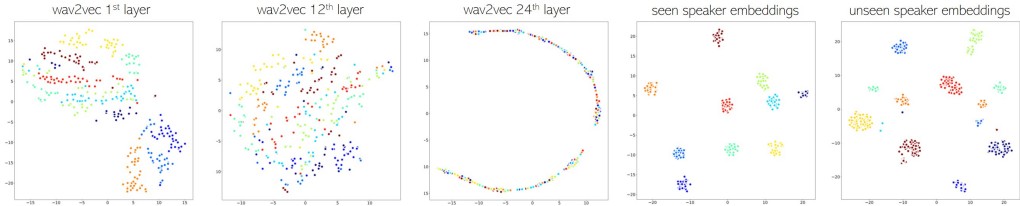

Figure 2: The visualization of intermediate representations of XLSR-53 using TSNE.

popular Yin algorithm to address this issue. The Yin algorithm uses the cumulative mean normalized difference function $d'_t(\tau)$ to extract frame-wise features from a raw waveform, which is defined as follows,

$$
d'_t(\tau) = \begin{cases} 1, & \text{if } \tau = 0 \\ d_t(\tau)/\sum_{j=1}^{\tau} d_t(j), & \text{otherwise.} \end{cases} \tag{1}
$$

The $d_t(\tau)$ is a difference function that outputs a small value when there exists a periodicity on time-lag $\tau$ and it is defined as follows,

$$
d_t(\tau) = \sum_{j=1}^{W} (x_j - x_{j+\tau})^2 = r_t(0) + r_{t+\tau}(0) - 2r_t(\tau), \tag{2}
$$

where $t$, $\tau$, $W$, and $r_t$ denote frame index, time lag, window size, and the auto-correlation function, respectively. After some post processing steps, the Yin algorithm selects $f_0$ from multiple $f_0$ candidates. See [13] for more details. Rather than explicitly selecting $f_0$, we would like to train the network to generate pitch harmonics from the output of the function $d'_t(\tau)$. However, $d'_t(\tau)$ itself is limited to be used as a pitch feature because it lacks controllability, unlike $f_0$. Therefore, we propose Yingram $Y$ by converting the time-lag axis to the midi-scale axis as follows,

$$
Y_t(m) = \frac{d'_t(\lceil c(m) \rceil) - d'_t(\lfloor c(m) \rfloor)}{\lceil c(m) \rceil - \lfloor c(m) \rfloor} \cdot (c(m) - \lfloor c(m) \rfloor) + d'_t(\lfloor c(m) \rfloor), \tag{3}
$$

$$
c(m) = \frac{sr}{440 \cdot 2^{(\frac{m-69}{12})}}, \tag{4}
$$

where $m$, $c(m)$, and $sr$ denote midi note, midi-to-lag conversion function, and sampling rate, respectively. We set 20 bins of Yingram to represent a semitone range. In addition, we set Yingram to represent the frequency between 10.77 hz and 1000.40 hz by setting $W$ to 2048 and the range of $\tau$ between 22 and 2047. In the training stage, the input to the synthesis network is the frequency range between 25.11 hz and 430.19 hz, which is shown as *scope* in Fig. 3. After the training is finished, we can change the pitch by shifting the scope. That is, in the inference stage, one could simply change the pitch of the speech signal by shifting the scope. For example, if we move the scope down 20 bins, the pitch can be raised by a semitone.

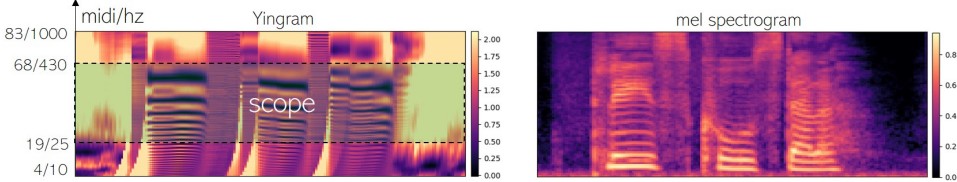

Figure 3: The visualization of Yingram and the corresponding mel spectrogram.

**Energy** For the energy feature, we simply took an average from a log-mel spectrogram along the frequency axis.

## 2.2 Synthesis Network

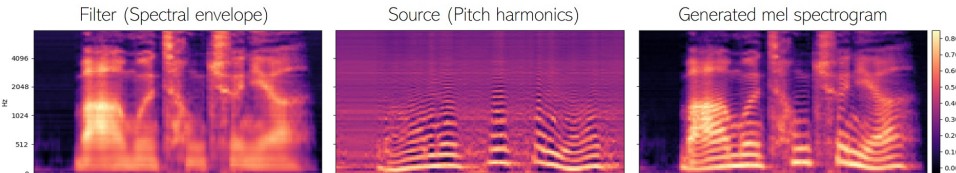

Figure 4: The outputs of $\mathcal{G}_S$ and $\mathcal{G}_F$. The two outputs from each generator are summed to reconstruct a mel spectrogram.

It is well-known that speech production can be explained by source-filter theory. Inspired by this, we separate synthesis networks into two parts, source generator $\mathcal{G}_S$ and filter generator $\mathcal{G}_F$. While the energy and speaker features are common inputs for both generators, $\mathcal{G}_S$ and $\mathcal{G}_F$ differ in that they take Yingram and wav2vec features, respectively. Because the acoustic feature can be interpreted as a sum of source and filter in the log magnitude domain, we incorporate inductive bias in the model by summing the outputs from each generator similarly to [25]. As will be discussed in more detail in the next section, even though the training loss is only defined using mel spectrograms, the network learns to separately generate the spectral envelope and pitch harmonics as shown in Fig. 4. Note that this separation not only provides the interpretability to the model but also enables formant preserving pitch shifting. To summarize, the acoustic feature, mel spectrogram $\hat{M}$, is generated as follows,

$$\hat{M} = \mathcal{G}_S(\text{Yingram}, S, E) + \mathcal{G}_F(\text{wav2vec}, S, E), \qquad (5)$$

where $S$ and $E$ denote speaker embedding and energy features. We used stacks of 1D-CNN layers with gated linear units (GLU) [12] for generators. The detailed neural architecture of the generator is described in Appendix B. Note that each generator shares the same neural architecture. The only difference is the input features to the networks. Finally, the generated mel spectrogram is converted to waveform using the pre-trained HiFi-GAN vocoder [24].

## 3 Training

### 3.1 Information Perturbation

In our initial experiments, a neural network can be easily trained to reconstruct mel spectrograms using only the wav2vec feature. This implies that the wav2vec feature contains not only rich linguistic information but also information related to pitch and speaker. For that reason, we would like to train $\mathcal{G}_F$ to selectively extract only the linguistic-related information from the wav2vec feature, not pitch and speaker information. In addition, we would like to train $\mathcal{G}_S$ to selectively extract only the pitch-related information from the Yingram feature, not speaker information. To this end, we propose to perturb the information included in input waveform $x$ by using three functions that are 1. formant shifting ($fs$), 2. pitch randomization ($pr$), and 3. random frequency shaping using a parametric equalizer ($peq$)[3]. We applied a function $f$ on the wav2vec input, which is a chain of all three functions as follows, $f(x) = fs(pr(peq(x)))$. On the Yingram side, we applied function $g$, which is a chain of two functions $fs$ and $peq$ so that $f_0$ information is still preserved as follows, $g(x) = fs(peq(x))$. This way, we expect $\mathcal{G}_F$ to take only the linguistic-related information from the wav2vec feature, and $\mathcal{G}_S$ to take only pitch-related from the Yingram feature. Since the wav2vec and Yingram features can no longer provide the speaker-related information, the control of speaker information becomes uniquely dependent on the speaker embedding. The overview of the information flow is shown in Fig. 1. The hyperparameters of the perturbation functions are described more in Appendix A.

### 3.2 Training Loss

We used L1 loss between the generated mel spectrogram $\hat{M}$ and ground truth mel spectrogram $M$ to train the generators and speaker embedding network. However, it is well-known that the speech synthesis networks trained with L1 or L2 loss suffer from over-smootheness of the generated acoustic feature, which results in poor quality of the speech signal. Therefore, in addition to the L1 loss, we used the recent speaker conditional generative adversarial training method to mitigate this issue [8]. Writing the discriminator as $\mathcal{D}(M, \boldsymbol{c}_+, \boldsymbol{c}_-) \coloneqq \sigma(h(M, \boldsymbol{c}_+, \boldsymbol{c}_-))$, Choi et al. [8] proposed to

---

[3]We used Parselmouth for $fs$ and $pr$ [20]. PEQ was implemented following [53].

use projection conditioning [27] not only with the positive pairs but also with the negative pairs as follows,

$$h(M, \boldsymbol{c}_+, \boldsymbol{c}_-) = \psi(\phi(M)) + \boldsymbol{c}_+^T \phi(M) - \boldsymbol{c}_-^T \phi(M), \tag{6}$$

where $M$ denotes a mel spectrogram, $\sigma(\cdot)$ denotes a sigmoid function, $\boldsymbol{c}_+$ denotes a speaker embedding from a positively paired input speech sample, and $\boldsymbol{c}_-$ denotes a speaker embedding from a randomly sampled speech utterance. $\phi(\cdot)$ denotes an output from the intermediate layer of discriminator and $\psi(\cdot)$ denotes a function that maps input vector to a scalar value. The detailed neural architecture of $\mathcal{D}$ is shown in Appendix B. The loss functions for discriminator $L_\mathcal{D}$ and generator $L_\mathcal{G}$ are as follows:

$$
\begin{aligned}
L_\mathcal{D} &= -\mathbb{E}_{(M, \boldsymbol{c}_+, \boldsymbol{c}_-) \sim p_{data}, \hat{M} \sim p_{gen}} [log(\sigma(h(M, \boldsymbol{c}_+, \boldsymbol{c}_-))) - log(\sigma(h(\hat{M}, \boldsymbol{c}_+, \boldsymbol{c}_-)))], \\
L_\mathcal{G} &= -\mathbb{E}_{(M, \boldsymbol{c}_+, \boldsymbol{c}_-) \sim p_{data}, \hat{M} \sim p_{gen}} [log(\sigma(h(\hat{M}, \boldsymbol{c}_+, \boldsymbol{c}_-)))] + |M - \hat{M}|.
\end{aligned}
\tag{7}
$$

### 3.3 Test-time Self-Adaptation

Although the synthesis network can reconstruct an intelligible speech from the wav2vec feature in most cases, we observed that the network sometimes outputs speech signals with wrong pronunciation, especially when tested on unseen languages. To alleviate this problem, we propose to modify only the input representation, that is, the wav2vec feature, without having to train the whole network again from scratch. As shown in Fig. 5, we first compute L1 loss between the generated mel spectrogram $\hat{M}$ and ground truth mel spectrogram $M$ in the test-time. Then, we

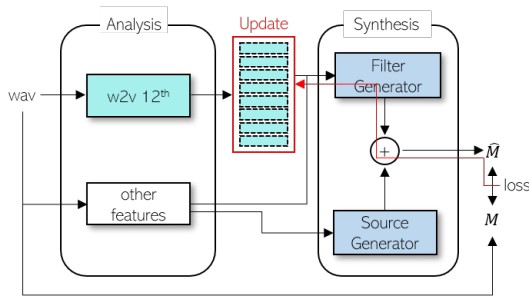

Figure 5: The illustration of TSA.

update only the parameterized wav2vec feature using the backpropagation signal from the loss. Note that the loss gradient (shown in red) is backpropagated only through the filter generator. Because this test-time training scheme requires only a single test-time sample and updates the input parameters by targeting the test-time sample itself, we call it test-time self-adaptation (TSA).

## 4 Experiments

### 4.1 Implementation Details

**Dataset**  To train NANSY on English, we used two datasets, i.e., 1. VCTK[4] [47], 2. train-clean-360 subset of LibriTTS[3] [54]. We trained the model using 90% of samples for each speaker. The speakers of train-clean-360 were included to the training set only when the total length of speech samples exceeds 15 minutes. To test on English speech samples we used two datasets; 1. For the *seen speaker* test we used *10% unseen utterances* of VCTK. 2. For the *unseen speaker* test we used *test-clean subset* of LibriTTS.

To train NANSY on multi-language, we used CSS10[3] dataset [32]. CSS10 includes 10 speakers and each speaker use different language. Note that there is *no* English speaking speaker included in CSS10. To train the model, we used 90% of samples for each speaker. To test on multilingual speech samples, we used the rest *10% unseen utterances* of CSS10.

**Training**  We used 22,050 hz sampling rate for every analysis feature except for wav2vec input that takes waveform with the sampling rate of 16,000 hz. We used 80 bands for mel spectrogram, where FFT, window, and hop size were set to 1024, 1024, and 256, respectively. The samples were randomly cropped approximately to 1.47-second, which results in 128 mel spectrogram frames. The networks were trained using Adam optimizer with $\beta_1 = 0.5$ and $\beta_2 = 0.9$. The learning rate was fixed to $10^{-4}$. We trained every model using one RTX 3090 with batch size 32. The training was done after 50 epochs.

---

[4]The licenses of the used datasets are as follows: 1. VCTK - Open Data Commons Attribution License 1.0, 2. LibriTTS - Creative Commons Attribution 4.0, and 3. CSS10 - Apache License 2.0.

## 4.2 Reconstruction

For the reconstruction (analysis and synthesis) tests, we report character error rate (CER (%)), and 5-scale mean opinon score (MOS ([1-5])), 5-scale degradation mean opinion score (DMOS ([1-5])). For MOS, higher is better. For DMOS and CER, lower is better. To estimate the characters from speech samples, we used google cloud ASR API. For MOS and DMOS, we used amazon mechanical turk (MTurk). The details of MOS and DMOS are shown in Appendix D.

**Yingram vs $f_0$** We compared two models trained with Yingram and $f_0$ to check which pitch feature shows more robust reconstruction performance. We used RAPT algorithm for $f_0$ estimation [42], which is known as a reliable $f_0$ tracker among many other algorithms [22]. Because RAPT algorithm works sufficiently well in most cases, we first manually listened to the reconstructed samples using the model trained with $f_0$. We first chose 30 reconstructed samples in the testset that failed to faithfully reconstruct the original samples using the model trained with $f_0$. After that we reconstructed the same 30 samples using the model trained with Yingram. Finally, we conducted ABX test to ask participants which of the two samples (A and B) sounds closer to the original sample (X). The ABX test was conducted on MTurk. The results showed that the participants chose Yingram with a chance of 68.3%. This shows that Yingram can be used as a more robust pitch feature than $f_0$, when $f_0$ cannot be accurately estimated.

**Reconstruction test** We randomly sampled 50 speech samples from VCTK (seen speaker) and sampled another 50 speech samples from *test-clean subset* of LibriTTS (unseen speaker) to test the reconstruction performance of NANSY trained on English datasets. The results in Table 1 shows that NANSY can perform high quality analysis and synthesis task. In addition, to test if the proposed framework can cover various languages, we trained and tested it with the multilingual dataset, CSS10. We randomly sampled 100 speech samples from CSS10 to test the reconstruction performance of NANSY trained on multi-language. The results are shown in Table 2. Although we do not impose any explicit labels for each language, the model was able to reconstruct various languages with high quality. The results of CER on each language is shown in Fig. 6, MUL.

|  | CER | MOS | DMOS |
|---|---|---|---|
| GT | n/a | $4.28 \pm 0.09$ | n/a |
| Recon | 5.6 | $4.18 \pm 0.09$ | $1.93 \pm 0.09$ |

Table 1: English reconstruction results.

|  | CER | MOS | DMOS |
|---|---|---|---|
| GT | n/a | $4.19 \pm 0.08$ | n/a |
| Recon | 7.3 | $4.14 \pm 0.09$ | $1.74 \pm 0.09$ |

Table 2: Multilingual reconstruction results.

**Test-time self-adaptation** To test the proposed test-time self-adaptation (TSA), we compared the CER performance of NANSY trained in three different configurations, 1. English (ENG), 2. English with TSA (ENG-TSA), 3. Multi-language (MUL). For every experiment, we iteratively updated the wav2vec feature 100 times. The results are shown in Fig. 6. Naturally, MUL generally showed better CER performance than other configurations. Interestingly, however, ENG-TSA sometimes showed similar or even better performance than MUL, which shows the effectiveness of the proposed TSA technique.

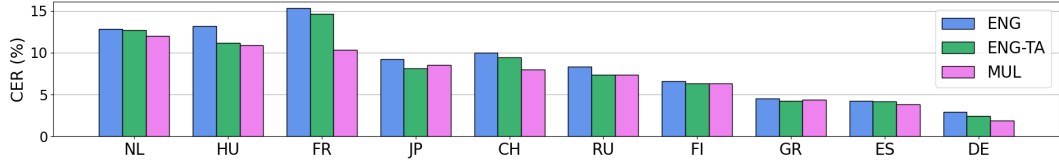

Figure 6: The CER results on 10 languages (NL: Dutch, HU: Hungarian, FR: French, JP: Japanese, CH: Chinese, RU: Russian, FI: Finnish, GR: Greek, ES: Spanish, DE: German).

## 4.3 Voice conversion

NANSY can perform zero-shot voice conversion by simply passing the desired target speech utterance to the speaker embedding network. We first compared the English voice conversion performance of NANSY with recently proposed zero-shot voice conversion models. Next, we tested the multilingual voice conversion performance. Finally, we tested unseen language voice conversion for both seen speaker and unseen speaker targets. Note that in every voice conversion experiment, we shifted the

median pitch of a source utterance to the median pitch of a target utterance by shifting the scope of Yingram. We measured naturalness with 5-scale mean opinion score (MOS [1-5]). Speaker similarity (SSIM (%)) were measured with a binary decision and uncertainty options, following [50]. For MOS and SSIM, we again used MTurk. The details of MOS and SSIM are shown in Appendix D. One of the crucial criteria for evaluating the quality of the converted samples is to check the intelligibility of them. Previous zero-shot voice conversion models, however, have only reported MOS or SSIM and have been negligent on assessing the intelligibility of the converted samples [36, 10, 51]. To this end, we report character error rate (CER (%)) between estimated characters of source and converted pairs. To estimate the characters from speech samples, we used google cloud ASR API.

**Algorithm comparison**  Here, we report three source-to-target speaker conversion settings, 1. seen-to-seen (many-to-many, M2M), 2. unseen-to-seen (any-to-many, A2M), 3. unseen-to-unseen (any-to-any, A2A). For every setting, we considered 4 gender-to-gender combination, i.e., male-to-male (m2m), male-to-female (m2f), female-to-male (f2m), and female-to-female (f2f). For M2M setting, we randomly sampled 25 seen speakers from VCTK and randomly assigned 2 random speakers from VCTK, resulting in 200 (=25×2×4) conversion pairs in total. For A2M setting, we randomly sampled 10 seen speakers from *test-clean subset* of LibriTTS and randomly assigned 2 random speakers from VCTK resulting in 80 (=10×2×4) conversion pairs in total. For A2A setting, we randomly sampled 10 seen speakers from *test-clean subset* of LibriTTS and randomly assigned 2 random speakers from *test-clean subset* of LibriTTS resulting in 80 (=10×2×4) conversion pairs in total. We trained three baseline models with official implementations - VQVC+ [51], AdaIN [10], AUTOVC [36] - using the same dataset and mel spectrogram configuration as NANSY. For a fair comparison, we used a pre-trained HiFi-GAN vocoder for every model. Table 3 shows that NANSY significantly outperforms previous models in terms of every evaluation measure. This implies that the proposed information perturbation approach does not suffer from the trade-off between CER and SSIM unlike information bottleneck approaches. The SSIM results for all possible gender-to-gender combinations are shown in Fig. 9 in Appendix C.

| | M2M | | | A2M | | | A2A | | |
|---|---|---|---|---|---|---|---|---|---|
| | CER[%] | MOS[1-5] | SSIM[%] | CER[%] | MOS[1-5] | SSIM[%] | CER[%] | MOS[1-5] | SSIM[%] |
| SRC as TGT | n/a | $4.23 \pm 0.05$ | 0 | n/a | $4.28 \pm 0.09$ | 0.60 | n/a | $4.26 \pm 0.07$ | 0.25 |
| TGT as TGT | n/a | $4.32 \pm 0.05$ | 94.9 | n/a | $4.29 \pm 0.05$ | 92.4 | n/a | $4.27 \pm 0.07$ | 96.2 |
| VQVC+ | 54.0 | $1.76 \pm 0.05$ | 54.5 | 74.7 | $1.73 \pm 0.11$ | 15.6 | 69.3 | $1.83 \pm 0.09$ | 13.8 |
| AdaIN | 62.9 | $2.22 \pm 0.07$ | 24.0 | 79.6 | $1.92 \pm 0.12$ | 18.1 | 59.3 | $2.12 \pm 0.10$ | 21.2 |
| AUTOVC | 31.7 | $3.41 \pm 0.06$ | 47.3 | 36.1 | $2.74 \pm 0.11$ | 33.2 | 28.2 | $2.59 \pm 0.08$ | 23.3 |
| NANSY | **7.5** | $\mathbf{3.79} \pm 0.07$ | **91.4** | **7.6** | $\mathbf{3.73} \pm 0.05$ | **88.1** | **8.6** | $\mathbf{3.44} \pm 0.07$ | **64.6** |

Table 3: Evaluation results on English voice conversion. SRC and TGT denote, source and target, respectively.

**Multilingual voice conversion**  We tested multilingual voice conversion performance with the model trained on the multilingual dataset, CSS10. We randomly sampled 50 samples for each language speaker from CSS10 and assigned random single target speaker from CSS10 for each source language, resulting in 500 (=50×10) conversion pairs in total. The results in Table 4 show that the proposed framework can successfully perform multilingual voice conversion by training it with the multilingual dataset. However, there is still a room for improvement for multilingual voice conversion when comparing to the results in Table 3, where NANSY is just trained on English.

**Unseen language voice conversion**  We tested the voice conversion performance on unseen source languages using NANSY trained on English. We tested the performance on two settings, 1. unseen source language (CSS10) to seen target voice (VCTK) and 2. unseen source language (CSS10) to unseen target voice (CSS10). For the first experiment, we randomly sampled 50 samples for each unseen language speaker from CSS10 and assigned random English target speakers from VCTK, resulting in 500 (=50×10) conversion pairs in total. The second experiment was conducted identically to the multilingual voice conversion experiment setting. The results in Table 5 shows that NANSY can be successfully extended even to unseen language sources, although there was a decrease on SSIM compared to the results in Table 4 (69.5% → 61.0%).

| | CER | MOS | SSIM |
|---|---|---|---|
| TGT as TGT | n/a | 4.23 ± 0.06 | 98.0 |
| NANSY | 18.8 | 3.68 ± 0.09 | 69.5 |

Table 4: The multilingual voice conversion results. The model was trained using only CSS10.

| | Seen Speaker | | | Unseen Speaker | | |
|---|---|---|---|---|---|---|
| | CER | MOS | SSIM | CER | MOS | SSIM |
| TGT as TGT | n/a | 4.24 ± 0.07 | 100 | n/a | 4.23 ± 0.06 | 92.0 |
| NANSY | 14.8 | 3.75 ± 0.10 | 90.0 | 15.6 | 3.76 ± 0.09 | 61.0 |

Table 5: The voice conversion results on unseen language dataset, CSS10. The model was trained using only the English datasets.

## 4.4 Pitch shift and time-scale modification

To check the robustness of pitch shift (PS) and time-scale modification (TSM) performance of NANSY, we compared it with other robust algorithms, i.e., PSOLA [5] and WORLD vocoder [29, 28].

**Pitch shift** We tested PS performance with 5 semitone ranges, -6, -3, 0, 3, 6. The pitch was changed by shifting the scope of the proposed Yingram feature. Note that '0' was used to check analysis-synthesis performance. We randomly selected 20 samples for each semitone range from *10% unseen utterances* of VCTK. We evaluated the naturalness of speech samples with MOS on MTurk. The results in Table 6 show that NANSY generally outperforms algorithms such as PSOLA and WORLD vocoder on PS task.

**Time-scale modification** We tested TSM performance with 5 time-scale ratios, 1/2, 1/1.5, 1, 1.5, 2. The time-scale was modified by simply manipulating the hop length of the analysis features. Note that '1' was used to check analysis-synthesis performance. We randomly selected 20 samples for each time-scale ratio from *10% unseen utterances* of VCTK. We evaluated the naturalness of speech samples with MOS on MTurk. The results in Table 7 show that NANSY achieved competitive performance on TSM compared to the well-established PSOLA and WORLD vocoder.

| | -6 | -3 | 0 | 3 | 6 |
|---|---|---|---|---|---|
| WORLD [28] | 3.43 | 3.53 | 3.85 | 3.63 | 3.53 |
| PSOLA [29] | **3.63** | 3.55 | 3.93 | 3.75 | 3.48 |
| NANSY | 3.60 | **3.68** | **4.05** | **3.90** | **3.78** |

Table 6: Pitch shift results.

| | 1/2 | 1/1.5 | 1 | 1.5 | 2 |
|---|---|---|---|---|---|
| WORLD [28] | 2.40 | 3.60 | 3.83 | 3.38 | **3.03** |
| PSOLA [29] | **2.55** | **3.66** | 3.95 | 3.68 | 2.85 |
| NANSY | 2.45 | 3.63 | **3.98** | **3.70** | 2.93 |

Table 7: Time-scale modification results.

## 5 Related Works

**Self-supervised representation learning and synthesis of speech** There has been an increasing interest in the self-supervised learning methods within the machine learning and speech processing community. Oord et al. [31] first proposed to use noise contrastive estimation loss to train speech representations. Baevski et al. [4] extended this idea by integrating masked language modeling [15]. Another popular self-supervised learning method for speech representation is to train a neural network by targeting multiple self-supervision tasks [34, 37]. Most recently, [35] used the discrete disentangled self-supervised representations to re-synthesize them into a waveform. Although using the discrete units has its own advantage in that it is disentangled with speaker information, we found that an inaccurate quantization process often leads to mispronounced samples, which is why we turned to use continuous representation as it can provide more accurate results on linguistic information.

**Zero-shot voice conversion** Research on zero-shot voice conversion has been most actively conducted through the information bottleneck approach. Qian et al. [36] proposed to perform zero-shot voice conversion by utilizing the pre-trained speaker recognition network and information bottleneck by carefully designing the bottleneck of an auto-encoder. Inspired by the success of style conversion in computer vision, Chou and Lee [10] also focused on restricting the information flow using instance normalization [44] and adaptive instance normalization [19]. Lastly, Wu et al. [51] used multiple vector quantization layers [30] to restrict the information flow.

---

[5] We used Parselmouth for PSOLA [20].

**Inductive bias for audio generation**    It has been shown that neural networks can be combined with traditional speech/sound production models for efficient and strong performance. One of the speech production models that has been integrated with neural networks is the source-filter model. By modeling source and filter components with deep networks, it has been used for applications such as vocoder [49, 23, 45] and acoustic feature generation [25]. Furthermore, Engel et al. [17] proposed to integrate a harmonic plus noise model [38] and neural networks to produce natural audio signal.

**Consistency learning**    Learning representations by augmenting the data has been one of the key ideas to leverage the performance of classification tasks [43, 52]. This shares the similar idea with the proposed information perturbation strategy in that the data is perturbed so that the neural network must learn to ignore the perturbations and learn the consistency from the data. However, *the key difference* between the consistency learning and the information perturbation is that the information perturbation method is designed for "generative" task and that it is the "decoder" (e.g., Generator) that is trained to selectively take the essential attributes to reconstruct the signal from the given perturbed representations.

## 6    Conclusions and Discussion

In this work, we proposed a neural analysis and synthesis framework (NANSY) that can perform zero-shot voice conversion, formant preserving pitch shift, and time-scale modification with a single model. The proposed model can be trained in a fully self-supervised manner, that is, it can be trained without any labeled data such as text or speaker information. We showed the effectiveness of the proposed information perturbation approach by showing the voice conversion results on various settings. Furthermore, we showed the effectiveness of the proposed TSA method by testing it on unseen languages, which shows the possibility of NANSY to be extended on low-resource languages. Although the proposed method empowers controllability over several attributes of a speech signal, it is still limited in terms of lacking controllability over linguistic information. As a future work, therefore, we would like to investigate on a hybrid approach that integrates text information as a side input so that the user can manipulate even the linguistic information in the speech signal. Finally, to prevent the proposed framework being used maliciously (e.g., voice phishing), it would be important to develop a detection algorithm that can discriminate a synthesized speech sample from a real speech sample. To examine the potential of such a detection system, we have tried using the trained Discriminator from the NANSY framework, which is expected to discriminate fake samples from real samples. We measured the accuracy of classifying 185 reconstructed samples and 185 ground truth samples. In addition, we measured the accuracy of classifying 560 voice conversion samples and 560 ground truth samples. The accuracy was 91.4% on the reconstruction set and 95.5% on the voice conversion set. This shows the possibility of Discriminator being used as a byproduct network to discriminate real speech samples from fake speech samples. However, we also found that Discriminator is prone to being deceived by the generated samples from other speech generative models as Discriminator was not jointly trained with those generative models. Therefore, we expect more robust synthesized speech detection algorithms to be developed in the future such as [48, 40, 9, 6].

## Broader Impacts

The proposed NANSY framework shows that a generative model can benefit from self-supervised representations. By choosing proper domain specific "information perturbation" functions, we believe that one can achieve controllable generative modeling in a fully self-supervised way. The information perturbation training strategy may also be used for other modalities and facilitate self-supervised representation learning methods too. With the proposed training framework, one can manipulate various aspects of speech samples. Among the various controllabilities, it is rather obvious that the voice conversion technique can be misused and potentially harm other people. More concretely, there are possible scenarios where it is being used by random unidentified users and contributing to spreading fake news. In addition, it can raise concerns about biometric security systems based on speech. To mitigate such issues, the proposed system should not be released without a consent so that it cannot be easily used by random users with malicious intentions. That being said, there is still a potential for this technology to be used by unidentified users. As a more solid solution, therefore, we believe a detection system that can discriminate between fake and real speech should be developed. The preliminary results of the detection system is reported in section 6.

## Acknowledgments and Disclosure of Funding

This work was supported by Institute of Information & communications Technology Planning & Evaluation (IITP) grant funded by the Korea government(MSIT) [NO.2021-0-01343, Artificial Intelligence Graduate School Program (Seoul National University)]

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
