# Appendix A  Information Perturbation

Here we describe the hyperparameters for three perturbation functions that are 1. formant shifting ($fs$), 2. pitch randomization ($pr$), and 3. random frequency shaping using parametric equalizer ($peq$).

For $fs$, a formant shifting ratio was sampled uniformly from $U(1, 1.4)$. After sampling the ratio, we again randomly decided whether to take the reciprocal of the sampled ratio or not.

For $pr$, a pitch shift ratio and pitch range ratio were sampled uniformly from $U(1, 2)$ and $U(1, 1.5)$, respectively. Again, we randomly decided whether to take the reciprocal of the sampled ratios or not. For more details for formant shifting and pitch randomization, please refer to Praat manual [5].

Lastly, $peq$ was used for random frequency shaping. $peq$ is a serial composition of low-shelving, peaking, and high-shelving filters. We used one low-shelving $H^{\text{LS}}$, one high-shelving $H^{\text{HS}}$, and eight peaking filters $H_1^{\text{Peak}}, \cdots, H_8^{\text{Peak}}$.

$$H^{\text{PEQ}}(z) = H^{\text{LS}}(z)H^{\text{HS}}(z)\prod_{i=1}^{8} H_i^{\text{Peak}}(z). \tag{8}$$

Each component is a second-order IIR filter and has a cutoff/center frequency, quality factor, and gain parameter. Cutoff frequencies of $H^{\text{LS}}$ and $H^{\text{HS}}$ were fixed to $60Hz$ and $10kHz$, respectively. Center frequencies of $H_1^{\text{Peak}}, \cdots, H_8^{\text{Peak}}$ were uniformly spaced in between the shelving filters on a logarithmic scale. We randomized the quality factor of each component to $Q = Q_{\min}(Q_{\max}/Q_{\min})^z$ where $Q_{\min} = 2$, $Q_{\max} = 5$, and $z \sim U(0, 1)$. We also randomized the gain (in decibel) of each component to $g \sim U(-12, 12)$. Refer to [53] for more details.

# Appendix B  Neural Architectures

**Generator**  The neural architecture of generator is shown in Fig. 7. Note that source generator ($\mathcal{G}_S$) and filter generator ($\mathcal{G}_F$) shares the same architecture. The only difference is the input to the network, where $\mathcal{G}_S$ takes (Yingram, energy, speaker embedding) features, and $\mathcal{G}_F$ takes (wav2vec, energy, speaker embedding) features. We used conditional layer normalization layer (cLN) for speaker conditioning following [7].

**Discriminator**  The neural architecture of discriminator is shown in Fig. 8. We used 1D-CNN-based residual blocks for Discriminator. The speaker embeddings from speaker network were used for conditioning.

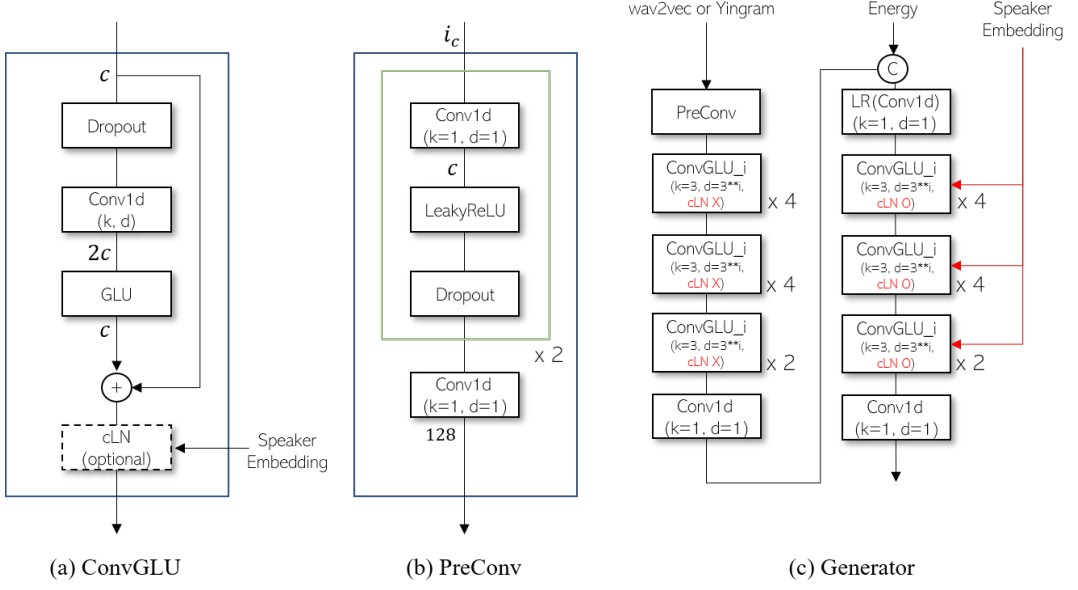

(a) ConvGLU      (b) PreConv      (c) Generator

Figure 7: The neural architecture of Generator. $k$, $d$, $c$, $i_c$, LR denotes kernel size, dilation, channel size, input channel size, and Leaky ReLU, respectively. The $i_c$ of $\mathcal{G}_S$ and $\mathcal{G}_F$ is 1024 (wav2vec) and 984 (Yingram), respectively.

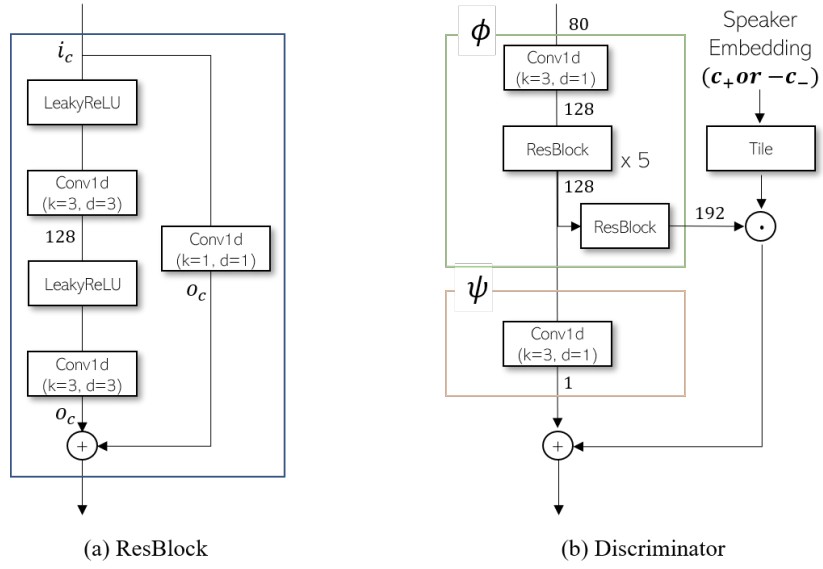

(a) ResBlock      (b) Discriminator

Figure 8: The neural architecture of Discriminator. $k$, $d$, $c$, $i_c$, $o_c$ denotes kernel size, dilation, channel size, input channel size, and output channel size, respectively. $c_+$ denotes a speaker embedding that is positively paired with an input speech signal and $c_-$ denotes a randomly sampled speaker embedding that is negatively paired with an input speech signal.

## Appendix C   Full Speaker Similarity Evaluation Results

The full evaluation SSIM results including all gender-to-gender combinations are shown in Fig. 9. The results shows that NANSY can achieve better voice conversion performance than baseline models in every gender-to-gender settings.

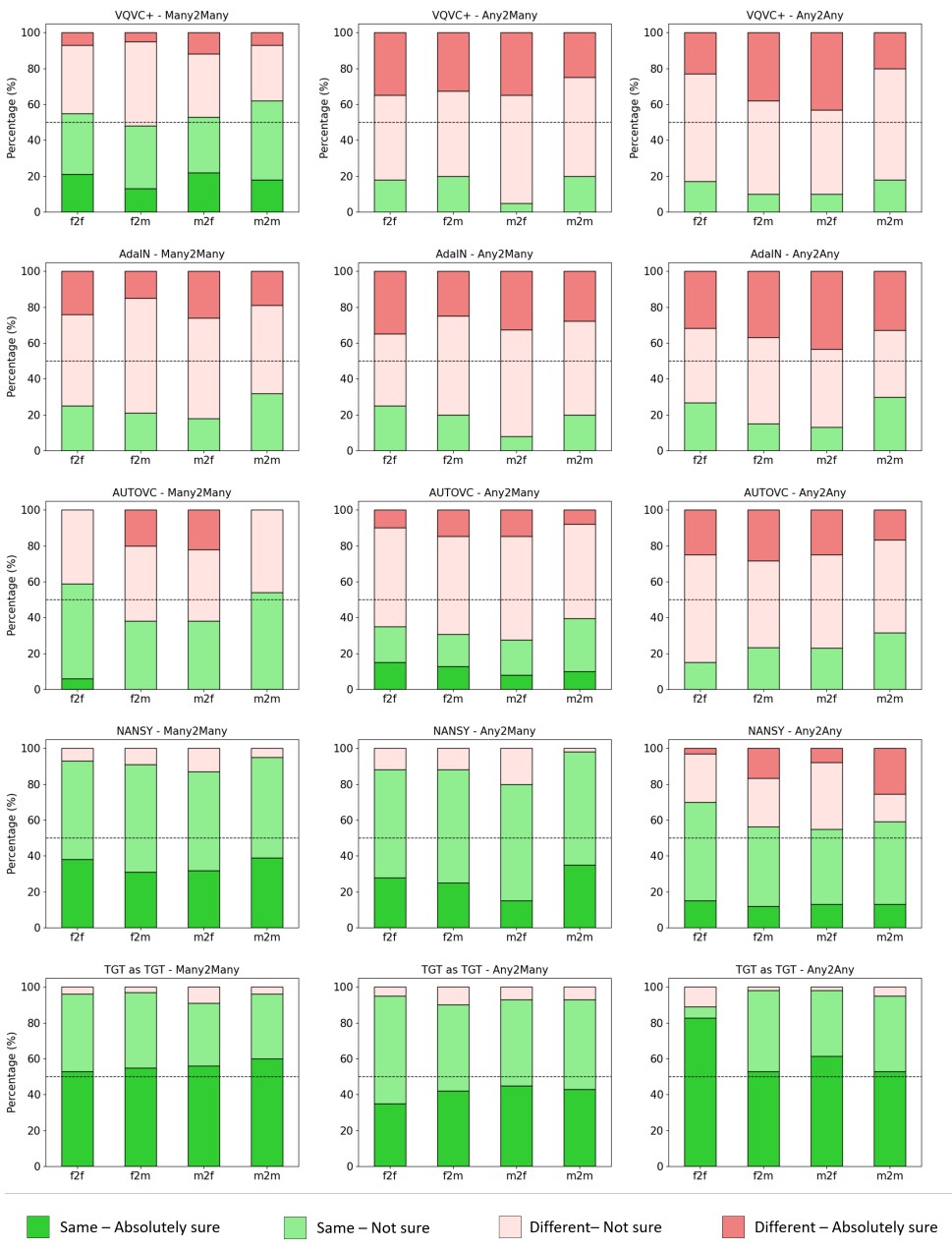

Figure 9: Full speaker similarity results. f2f: female-to-female. f2m: female-to-male. m2f: male-to-female. m2m: male-to-male.

# Appendix D    Crowdsource Evaluation

Here we attach the webpage instructions for 4 crowdsource evaluations based on MTurk. The instructions for mean opinion score (MOS), degradation mean opinion score (DMOS), speaker similarity (SSIM), and ABX test are shown in Fig. 10, Fig. 11, Fig. 12, and Fig. 13, respectively.

The task unit of MTurk is called Human Intelligence Task (HIT).

For MOS, the total amount of HITs were 6108. Each HIT was assigned to 2 subjects. The reward for each HIT was 0.05 USD. The total amount of budget we spent for MOS was therefore, 716 USD. The estimated hourly wage for MOS is 18 USD.

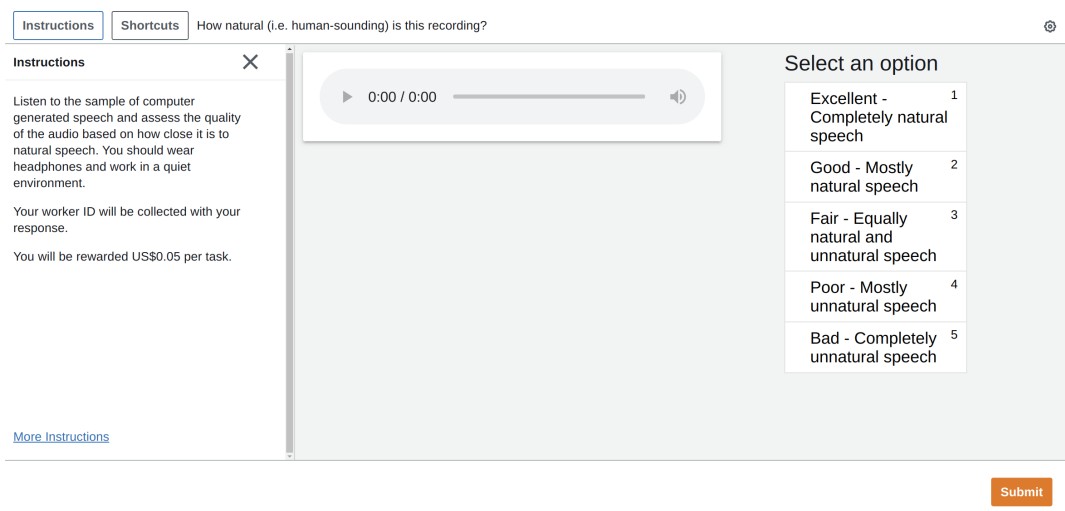

Figure 10: Mean opinion score test instruction.

For DMOS, the total amount of HITs were 200. Each HIT was assigned to 2 subjects. The reward for each HIT was 0.1 USD. The total amount of budget we spent for MOS was therefore, 40 USD. The estimated hourly wage for DMOS is 18 USD.

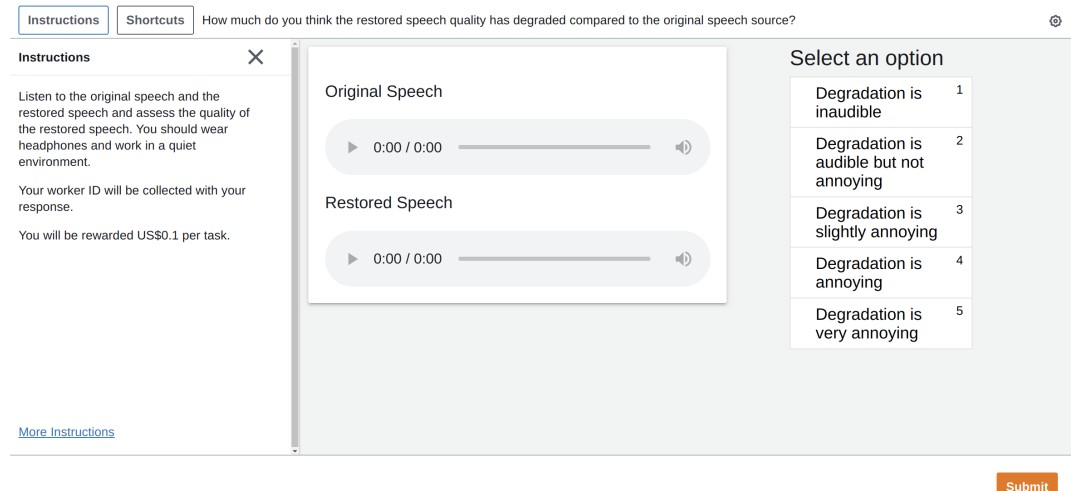

Figure 11: Degradation mean opinion score test instruction.

For SSIM, the total amount of HITs were 5160. Each HIT was assigned to 2 subjects. The reward for each HIT was 0.1 USD. The total amount of budget we spent for SSIM was therefore, 1032 USD. The estimated hourly wage for SSIM is 18 USD.

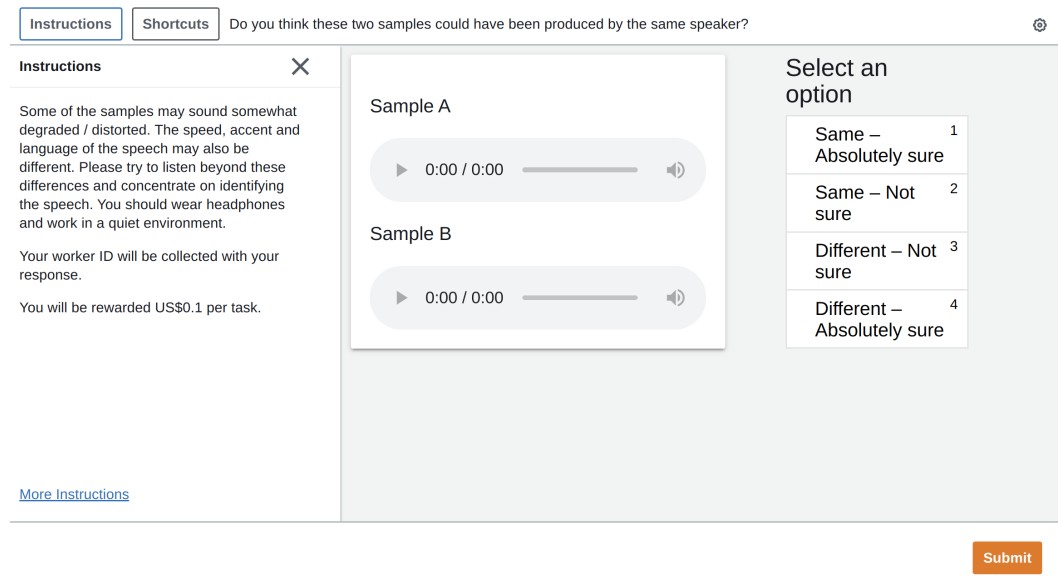

Figure 12: Speaker similarity test instruction.

For ABX, the total amount of HITs were 150. Each HIT was assigned to 5 subjects. The reward for each HIT was 0.1 USD. The total amount of budget we spent for SSIM was therefore, 75 USD. The estimated hourly wage for ABX is 12 USD.

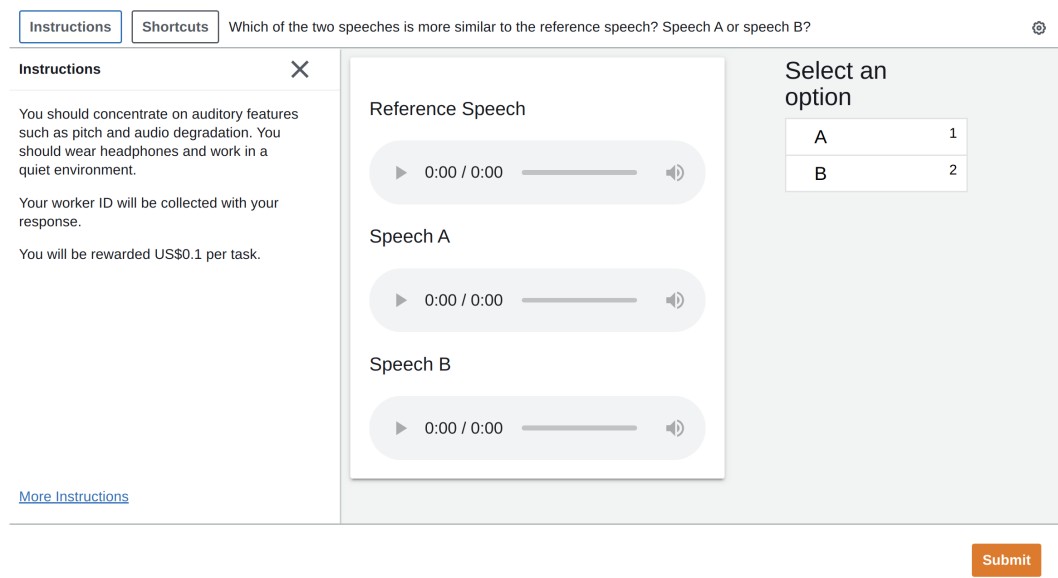

Figure 13: ABX test instruction.