# OpenReview forum: "Neural Analysis and Synthesis: Reconstructing Speech from Self-Supervised Representations"
_NeurIPS.cc/2021/Conference — NeurIPS 2021 Poster_

### Official Review · Reviewer_C3ov · 2021-07-06

**Rating:** 7
**Confidence:** 4

**Summary:**

The paper proposes a speech analysis/synthesis framework that allows users to manipulate speech traits such as pitch, identity, and time scale. The framework exploits existing self-supervised representations for speaker identity and content, and develops a new one for pitch (Yingram). Results are shown for speech reconstruction, voice conversion, pitch shift, and timescale adaptation. Emphasis is put on multilingual and zero-shot scenarios.

**Ethical Concerns:**

I could not spot any ethical issue.


**Limitations And Societal Impact:**

Yes, this is done in the conclusions section.

**Main Review:**

The paper is well-written and shows a thorough evaluation on different but related tasks that I think do a good job in convincing the reader that their method works. The audio samples provided are impressive, which made me increase my review score. The fact that existing self-supervised representations can be used to re-synthesize speech is far from clear (and, to me, rather unintuitive), and the paper could be considered a first work in showing that.

Originality --- The paper borrows from existing work to construct a system that, although not very original, works well in terms of performance.

Quality --- Given the evaluations done and the audio samples provided, my impression is that the system performs quite well. I'd have liked to see some intermediate evaluations (see below) but the whole result is ok.

Clarity --- The explanations are clear although they could be improved a bit (see below for some suggestions).

Significance --- I think that the paper will hardly inspire other machine learning disciplines. However, I think it sets a bar for the sub-topic of speech modeling and synthesis. The focus on multilingual and zero-shot scenarios is of relevance. There is also the aforementioned point of showing that pre-trained self-supervised representations can be exploited to synthesize speech with good results.

Additional comments:
* I do not very much agree with the classification between "text-based" and "information bottleneck" approaches. I find that there are many approaches, specially in voice conversion (VC), that do not fall within these two categories and that, therefore, such classification is too much of a reductionist one. A couple of examples of such VC approaches that the authors could consider citing from past NeurIPS are [A,B].
* I think it would be much clear if authors showed the losses they use in Fig 1. Do I understand correctly that, if the vocoder is pre-trained, all losses are applied before the vocoder?
* I think more detail could be provided in the "Pitch" section of page 4. Specially for the proposed new feature. It could be also interesting that the authors shared their code for this part in the future, to foster further re-use.
* In Line 171, it is mentioned that a pre-trained HiFi-GAN vocoder is used, but this somehow seems to conflict with the multi-language setting right? (was HiFi-GAN trained with multiple languages or shown to perform well in such scenario?) It would have been interesting to comment that and perhaps include some experiment to assess the "effect" of using this pre-trained GAN (and not another one) in their approach. Which is the impact? How much success does depend on the vocoder? ... Which would be the MOS of HiFiGAN alone in Tables 1-2? Which would be the MOS of the same system with another vocoder?
* Similarly for the "information perturbation" approach. Which is the effect of not using it? Could the authors provide any quantitative measurement? In general, I find that the paper lacks some ablation or quantitative assessment of the different proposals. It would be good to provide such assessment to try to elucidate which are the crucial/critical components of the system.
* Please do not use more than one character for mathematical variables and functions. For instance, $fs$ in mathematical notation means $f$ multiplied by $s$. Also, in that case, $f$ is also confusing with function $f$ (Sec 4.1). I consider using more than one character simply bad practice and should be avoided at all cost in papers for renowned conferences like NeurIPS.
* For the VC case, while reading Sec 5.3, I was a bit confused on how the identity of the target speaker was extracted. Perhaps this part can be improved (it helped me quite a bit the diagrams shown in the demo/samples page).

[A] https://papers.nips.cc/paper/2018/hash/4559912e7a94a9c32b09d894f2bc3c82-Abstract.html

[B] https://proceedings.neurips.cc/paper/2019/hash/9426c311e76888b3b2368150cd05f362-Abstract.html




**Time Spent Reviewing:**

3

---

> ### Author Response · Authors · 2021-08-09
> **Response to Reviewer C3ov**
>
> We thank reviewer_C3ov for the constructive and detailed comments.
> Below are the responses to each of your concerns.
>
>
> \
> I do not very much agree with the classification between "text-based" and "information bottleneck" approaches. I find that there are many approaches, specially in voice conversion (VC), that do not fall within these two categories and that, therefore, such classification is too much of a reductionist one. A couple of examples of such VC approaches that the authors could consider citing from past NeurIPS are [A,B].
>
> **Authors’ response**: We agree with the reviewer’s opinion since there exist numerous voice conversion papers out there, and some of them are hard to categorize in one of the two categories we introduced in our manuscript. We will try to revise this part by citing the papers from past NeurIPS.
>
> \
> I think it would be much clear if authors showed the losses they use in Fig 1. Do I understand correctly that, if the vocoder is pre-trained, all losses are applied before the vocoder?
>
> **Authors’ response**: Thank you for your constructive suggestion. We did not put it in the figure because we thought it might deter the readability if it contained too much information in a single figure. But yes, you understood it correctly.
>
> \
> I think more detail could be provided in the "Pitch" section of page 4. Specially for the proposed new feature. It could be also interesting that the authors shared their code for this part in the future, to foster further re-use.
>
> **Authors’ response**: Thank you for your suggestion. After internal discussions, following the reviewer’s suggestion, we decided to release the code for Yingram publicly.
>
> \
> In Line 171, it is mentioned that a pre-trained HiFi-GAN vocoder is used, but this somehow seems to conflict with the multi-language setting right? (was HiFi-GAN trained with multiple languages or shown to perform well in such scenario?) It would have been interesting to comment that and perhaps include some experiment to assess the "effect" of using this pre-trained GAN (and not another one) in their approach. Which is the impact? How much success does depend on the vocoder? ... Which would be the MOS of HiFiGAN alone in Tables 1-2? Which would be the MOS of the same system with another vocoder?
>
> **Authors’ response**: Surprisingly, we found that pretrained HiFi-GAN works very well in multiple languages. We used the official pretrained Universal HiFi-GAN V1 from the HiFi-GAN github repository (https://github.com/jik876/hifi-gan). Therefore, no further training was done regarding the vocoder part. The same HiFi-GAN vocoder was used for every model for a fair comparison to show that the success of our model is not dependent on HiFi-GAN, as we have stated in line 300-301. HiFi-GAN vocoder can generate a realistic waveform when given a mel spectrogram that is close to the ground truth mel spectrogram. Therefore, the quality of the final speech sample is guaranteed if the acoustic model can produce mel spectrograms as realistic as possible.
>
> \
> Similarly for the "information perturbation" approach. Which is the effect of not using it? Could the authors provide any quantitative measurement? In general, I find that the paper lacks some ablation or quantitative assessment of the different proposals. It would be good to provide such assessment to try to elucidate which are the crucial/critical components of the system.
>
> **Authors’ response**: In our initial experimental phase, we actually tried to conduct an ablation study to compare the model trained without information perturbation. However, the model almost completely failed to perform voice conversion without information perturbation. This is actually an expected behavior as wav2vec features contain enough information (e.g., linguistic, timbre and pitch) to reconstruct the mel spectrogram, which we have experimented and stated in line 174-176. wav2vec features without information perturbation have richer frame-wise fine-grained information than the time-pooled speaker representation from speaker embedding network. We found that this causes the generator network to simply ignore the provided information from a speaker embedding, which is also sometimes called posterior collapse in probabilistic point of view. Therefore, we thought it was sort of meaningless and a waste of money to conduct an extra ablation study for it when it is too obvious. But we agree with the reviewer’s opinion to question the effect of information perturbation . We will try to note this more specifically in our revised manuscript.
>
> \
> Please do not use more than one character for mathematical variables and functions. For instance, fs  in mathematical notation means f multiplied by s. Also, in that case, f is also confusing with function f (Sec 4.1). I consider using more than one character simply bad practice and should be avoided at all cost in papers for renowned conferences like NeurIPS.
>
> **Authors’ response**: We thank you for your comments. We will change the notations of functions by representing them with one character so that there will be no further confusion.
>
> \
> For the VC case, while reading Sec 5.3, I was a bit confused on how the identity of the target speaker was extracted. Perhaps this part can be improved (it helped me quite a bit the diagrams shown in the demo/samples page).
>
> **Authors’ response**: Thank you for your suggestion. We will try to add the figures and diagrams from the demo page to Appendix..

---

### Official Review · Reviewer_DLtc · 2021-07-14

**Rating:** 6
**Confidence:** 4

**Summary:**

The paper attempts to separate speech into four stream of features, the phonetic content, the speaker, the energy, and the fundamental frequency. The four stream are then reassembled back into the original input speech. Some of the streams, namely the energy and the fundamental frequency, are more straightforward to extract. While the others, namely the phonetic content and the speaker, rely the use of wav2vec 2.0. The separation of information is further encouraged by adding perturbation, as an approach to corrupt the information in one stream encouraging the model to pay attention to the other streams. Experiments are evaluated based on the quality of reconstruction, the ability to do voice conversion while holding certain property of the speech fixed.

**Limitations And Societal Impact:**

Voice conversion can be particularly concerning for its potential negative uses (such as identity theft) and for its bias on underrepresented groups.

**Main Review:**

I consider this as a system building paper, and give the paper a score 6. The novelty is in the combination of existing techniques, and the experiments are mainly about comparing surface performance, such as CER and MOS, and not about revealing properties of the proposed model. The presentation has room for improvement, particularly for the fragmented sections. The approach relies on many assumptions and has many limitations, and those should be made clear in the paper.

# Novelty

The paper's novelty is slightly lacking, because most of the ideas, even the combination of some, have already been proposed.

The idea of separating speech into multiple (more or less) independent streams has been explored, for example in [33], as the paper openly acknowledged. The use of wav2vec features for synthesis are also studied, particularly in the ZeroSpeech challenge. The use of perturbation to control or even to shift the attention of the model on certain streams is also used in the past, again in [33] for example.

The training loss is largely borrowed from [7], and the test-time adaptation has also been studied in fine-tuning large pre-trained models.

That said, as a system building paper, the particular combination is still interesting.


# Approach

The major assumption is that speech can actually be separated into four, almost independent, streams. This is largely false. For example, duration, a major aspect of prosody, is missing in the picture. The duration information is probably spread across the wav2vec, the energy, and the fundamental frequency streams. Another example is accent. Certain words are pronounced in certain ways depending on the speaker, so the phonetic features in wav2vec might contain the speakers' preference.

The perturbation approach takes advantage of that certain properties in speech are easy to manipulate. This is a source of prior knowledge, and the limitation is the approach can only be applied to things that we can manipulate. The perturbation approach also does not help solve the problems regarding duration and accents.

The reliance on wav2vec is reasonable, but at the same time a little concerning. The design of wav2vec is to encourage the learning of phonetic content. We do not know for certain that the model is going to learn the phonetic content, though a lot of design effort has been put in to make this happen. I am not entirely sure how much the proposed model depends on the first-layer wav2vec features being speaker specific. We do not know what promotes wav2vec to learn speaker information at the first layer, and we also do not know whether the behavior is reliable (for example, when we change the hyperparameters or the data that is trained on).


# Presentation

The presentation has some room for improvement.

The paper in general is too fragmented, that is, having many isolated subsections and paragraphs without discussing them together. Section 2, 3, 4, 5 are particularly fragmented. It would be great to have the following.

* a brief overview of the model and training in the intro
* a paragraph in section 2 about the landscape of the related work
* a paragraph in section 3 about the models at a high level (such as the captions in Figure 1)
* a paragraph in section 3 about how the individual components interact with each other
* a paragraph in section 4 about how the training interacts with the model design
* a paragraph in section 5 about the overall goal of the experiments

The related work section comes too soon. It's hard to contrast with the prior art without even understanding what is being proposed. In fact, the related work section in the current form does not put the proposed model in context and does not make any comparison.

The figures really help improve the understanding.


# Experiments

The experiments further solidify that this is a system building paper. All experiments are about comparing to others at the high level with surface metrics, such as CER, MOS, and speaker similarity. It is hard to draw any conclusion beyond that the system works to better than the ones compared. It would be better to have experiments that support or reject hypotheses, revealing properties of the models.


**Time Spent Reviewing:**

6

---

> ### Author Response · Authors · 2021-08-09
> **Response to Reviewer DLtc**
>
> We thank reviewer_DLtc for the insightful comments.
> Below are the responses to each of your concerns.
>
>
> \
> Novelty \
> The paper's novelty is slightly lacking, because most of the ideas, even the combination of some, have already been proposed.
> The idea of separating speech into multiple (more or less) independent streams has been explored, for example in [33], as the paper openly acknowledged. The use of wav2vec features for synthesis are also studied, particularly in the ZeroSpeech challenge. The use of perturbation to control or even to shift the attention of the model on certain streams is also used in the past, again in [33] for example.
> The training loss is largely borrowed from [7], and the test-time adaptation has also been studied in fine-tuning large pre-trained models.
> That said, as a system building paper, the particular combination is still interesting.
>
> **Authors’ response**: Thank you for your detailed comments. It is true that separating speech into multiple independent streams has been explored in previous works. The AUTO-VC [33] work is indeed one of the pioneering papers in the field of voice conversion. But to our best knowledge, there have been no previous works that achieve controllability in four aspects with a single model (e.g., pitch, speed, loudness, timbre). Especially, we achieve this in a fully-self-supervised way, which we believe deserves novelty.
> We are not sure which part of [33] attempted an information perturbation strategy. As far as we know [33] proposed an information "bottleneck" strategy that is responsible for the speech samples of relatively poor intelligibility.
> We agree that the training loss is borrowed from [7] and do not take it as one of our contributions.
> The test-time adaptation itself has been used in some classification studies or fine-tuning large pre-trained models. But it is new in that we are using it for a generation task. In addition, it is different in that we do not fine-tune the network but the feature itself, which makes the fine-tuning procedure more efficient.
>
> \
> The major assumption is that speech can actually be separated into four, almost independent, streams. This is largely false. For example, duration, a major aspect of prosody, is missing in the picture. The duration information is probably spread across the wav2vec, the energy, and the fundamental frequency streams. Another example is accent. Certain words are pronounced in certain ways depending on the speaker, so the phonetic features in wav2vec might contain the speakers' preference.
>
> **Authors’ response**: We deeply agree with the reviewer’s point of view. The voice identity of a speaker is actually very complex and can be viewed in a multifaceted way (e.g., prosody, velocity, accent, etc.). In most of the recent voice conversion papers, however, the speaker information is typically considered “time-invariant timbre”, which is actually not true. Therefore, it is indeed something we are aiming to study as our future work.
> In this work, however, following previous assumptions in many papers, we aimed to disentangle the controllable factors into four. In addition, disentangling only the time-invariant part of speech is sometimes practically necessary. For example, it is required in content dubbing applications where one would like to keep the original acting source but change only the time-invariant characteristics of the target speaker (as provided in our audio demo samples).
>
> \
> The perturbation approach takes advantage of that certain properties in speech are easy to manipulate. This is a source of prior knowledge, and the limitation is the approach can only be applied to things that we can manipulate. The perturbation approach also does not help solve the problems regarding duration and accents.
>
> **Authors’ response**: We also agree with this point. The perturbation approach can only be applied to things that we can manipulate, therefore, not a universal approach. Conversely, however, that also means that the perturbation approach can take advantage of many properties that we can manipulate at signal level. We believe there are still many properties left we can try to control at signal level to achieve more controllability. The perturbation approach might not be able to perturb the accent information, but we can also perturb duration using a time stretch function. Therefore, we believe the range of perturbation functions are actually wider than the reviewer might think. In addition, we don’t think the use of prior knowledge is something that should be discouraged. Each domain has its own source of prior knowledge. For example, in the field of computer vision, the easiest example would be to shuffle rgb channels or shift the coordinates of objects. The nice part of the information perturbation method is that the perturbation function does not have to be very accurate. It is enough if the function can just leave the semantic part one would like to take control of. Therefore, the idea can be easily and widely adopted in many other fields using their own source or prior knowledge. Lastly, the advantage of the perturbation approach is that it can be easily combined with previous methods (or maybe the future methods that will be proposed) as it is orthogonal to them (e.g., information bottleneck approaches).
>
> \
> The reliance on wav2vec is reasonable, but at the same time a little concerning. The design of wav2vec is to encourage the learning of phonetic content. We do not know for certain that the model is going to learn the phonetic content, though a lot of design effort has been put in to make this happen. I am not entirely sure how much the proposed model depends on the first-layer wav2vec features being speaker specific. We do not know what promotes wav2vec to learn speaker information at the first layer, and we also do not know whether the behavior is reliable (for example, when we change the hyperparameters or the data that is trained on).
>
> **Authors’ response**: We agree with the reviewer that the reliance on wav2vec can be a little concerning. Of course it would be better if we can train this system without wa2vec features. We are also working on training NANSY without the wav2vec feature. In our preliminary study, we found that we can train NANSY only using mel spectrograms with careful setting but we are still working on it. Beyond that, we are also working on new self-supervised training methods specifically designed for controllable generative tasks, which is again achieved using information perturbation.
>
> \
> The presentation has some room for improvement.
> The paper in general is too fragmented, that is, having many isolated subsections and paragraphs without discussing them together. Section 2, 3, 4, 5 are particularly fragmented. It would be great to have the following.
> * a brief overview of the model and training in the intro
> * a paragraph in section 2 about the landscape of the related work
> * a paragraph in section 3 about the models at a high level (such as the captions in Figure 1)
> * a paragraph in section 3 about how the individual components interact with each other
> * a paragraph in section 4 about how the training interacts with the model design
> * a paragraph in section 5 about the overall goal of the experiments
> The related work section comes too soon. It's hard to contrast with the prior art without even understanding what is being proposed. In fact, the related work section in the current form does not put the proposed model in context and does not make any comparison.
> The figures really help improve the understanding.
>
> **Authors’ response**: Thank you very much for your constructive suggestions. We will try to integrate the ideas to make better paper in the next version of our manuscript. In particular, we will try to edit the related work section reflecting the suggestions.

---

### Official Review · Reviewer_85wX · 2021-07-15

**Rating:** 7
**Confidence:** 4

**Summary:**

This paper presents a framework for speech conversion using self-supervised representations. The framework, called NANSY, consists of two parts: analysis and synthesis. A wav2vec pre-trained model is used to extract speech representation to train the filter generator in the synthesis module. Similarly, the wav2vec features from the first layer is used to estimate a speaker embedding network. The speaker embeddings are used as inputs to both the source and filter generators.   A novel Yingram feature is proposed to train the source generator. Yingram is derived by mapping the difference function from the Yin algorithm (well-known for pitch estimation) from the time-lag axis to the midi-scale axis. The outputs from the analysis module can be controlled to manipulate the output speech (useful for voice conversion and time/pitch scale modifications). The paper also proposed a method for test-time self-adaptation by updating the wav2vec representation features using the L1 loss between the original and reconstructed spectrograms.

The effectiveness of the proposed framework was evaluated on several tasks using a combination of the VCTK, LibriTTS and CSS10 multilingual datasets:
- speech reconstruction:
  - the speech reconstructed using Yingram is preferred over f0 in a subjective A-B test.
  - the reconstructed speech has similar MOS scores compared to the original speech.
  - test-time self-adaptation can achieve slightly better character error rate (CER) performance when the reconstructed speech is used in a downstream ASR task.
- Voice conversion:
  - Compared NANSY with several existing methods (VQVC+, AddIN and AUTOVC) and showed that NANSY can achieved significantly better CER performance, naturalness MOS and speaker similarity.
  - NANSY works for multilingual and unseen language voice conversion.
- Pitch and time modifications:
 - NANSY achieved better pitch shift modification performance compared to WORLD and PSOLA.
 - NANSY achieved similar time scale modification performance compared to WORLD and PSOLA.



**Limitations And Societal Impact:**

There is lack of ablation studies to understand the importance of speech perturbation on reconstruction quality. There are also no comparison with other techniques for the results in Tables 4 and 5.

**Main Review:**

Originality:

The main novelty of the proposed NANCY framework is the use of the wav2vec speech representation to avoid having a bottleneck structure (used in existing voice conversion work to have better controllability) and therefore can achieve good quality  reconstruction. The Yingram feature is proposed to achieve better control for pitch modification. Speech perturbation is used to encourage the generators in the synthesis module to select the relevant attributes.

Quality:

Overall, the design of the NANSY framework is well-motivated and it appears to achieve good performance improvements for voice conversion. The results in table 3 show that NANSY is substantially better compared to other methods, especially for the CER and SSIM metrics. Please provide explanation for the large improvements observed. It is also important to clarify that NANSY uses the pre-trained wav2vec model, which has been trained large amount of speech data that were not used in the other models.

Clarity:

Overall, the paper is well written and easy to follow. However, there are several aspects that need clarification:
- For the Yingram vs f0 comparison in Section 5.2, it seems biased if the ABX test was done based on the failure cases using f0. It would have been better to select half of the samples based on the failure cases using Yingram.
- For the test-time self-adaptation, it seems to assume that the ground truth spectogram is available to compute the L1 loss. How could this be useful in practice when ground truth is not available at test time.
- For the speaker embedding network, it is unclear how it is trained without speaker labels. It is important to provide more traning details.

Some minor comments:
- Line 288: "google" $\rightarrow$ "Google".
- Line 311: "there is still a room" $\rightarrow$ "there is still room".
- In various tables, the CER results for the ground truth or target are n/a. Why are they not available?

Significance:

The proposed framework leverages on the recent advancement in speech representation learning to achieve high quality speech reconstruction and voice conversion. In particular, the improvement in voice conversion quality is quite substantial.


**Time Spent Reviewing:**

2.5

---

> ### Author Response · Authors · 2021-08-09
> **Response to Reviewer 85wX**
>
> We thank reviewer_85wX for the constructive and detailed comments.
> Below are the responses to each of your concerns.
>
>
> \
> Please provide explanation for the large improvements observed. It is also important to clarify that NANSY uses the pre-trained wav2vec model, which has been trained large amount of speech data that were not used in the other models.
>
> **Authors’ response**: NANSY benefits from both information perturbation and the wav2vec feature. In our preliminary study, we found that we can still achieve adequate performance by replacing wav2vec features with mel spectrograms. However, the training was unstable and sometimes the training failed with little changes in configurations. The intelligibility of output speech samples were also a bit degraded when compared to using wav2vec features but still better than baseline models.
> We also found that AUTOVC-like information bottleneck almost always results in not very intelligible speech samples even combined with wav2vec features. Therefore, we do not think using the wav2vec feature is a sole source of performance improvement.
>
>
> \
> Clarity: Overall, the paper is well written and easy to follow. However, there are several aspects that need clarification
>
> For the Yingram vs f0 comparison in Section 5.2, it seems biased if the ABX test was done based on the failure cases using f0. It would have been better to select half of the samples based on the failure cases using Yingram.
>
> **Authors’ response**: We agree with this point and we actually tried to find samples where the model trained with Yingram fails to reconstruct the original input samples. However, there were little to no samples that failed to reconstruct the original samples.
>
> \
> For the test-time self-adaptation, it seems to assume that the ground truth spectogram is available to compute the L1 loss. How could this be useful in practice when ground truth is not available at test time.
>
> **Authors’ response**: Considering the use case of NANSY, there are no applications where ground truth source is not given. NANSY is about editing original speech source in a desirable manner, hence a speech sample is assumed to be given in any cases. For example, in voice conversion, there are always sources and targets. One can always fine-tune the wav2vec feature with the source sample and then convert the voice using the target sample.
>
> \
> For the speaker embedding network, it is unclear how it is trained without speaker labels. It is important to provide more traning details.
>
> **Authors’ response**: Because of the information perturbation function, the only feature that contains speaker timbre information is the speaker embedding among every analysis feature. Therefore, in order to reconstruct the mel spectrogram, which is forced by L1 loss, the speaker network must capture the speaker information.
>
> \
> Some minor comments:
> * Line 288: "google" → "Google".
> * Line 311: "there is still a room" →"there is still room".
> * In various tables, the CER results for the ground truth or target are n/a. Why are they not available?
>
> **Authors’ response**: Thank you for your detailed comments. We will edit our manuscript reflecting all typos. The CER results for the ground truth were denoted N/A as the CER was measured between the output of GoogleAPI(GroundTruthSample) and GoogleAPI(ReconstructedSample). This is because our goal is not to check how well the network has faithfully reconstructed a speech sample from ground truth text, but how similarly the network has reconstructed a sample to the original ground truth speech sample. In addition, we have also found some errors in the ground truth text annotations in the CSS10 dataset, therefore, we decided to leave it as N/A for consistency.
>
> \
> There is lack of ablation studies to understand the importance of speech perturbation on reconstruction quality. There are also no comparison with other techniques for the results in Tables 4 and 5.
>
> **Authors’ response**: We did not perform ablation study without information perturbation because the training simply fails to perform voice conversion without it. We will try to note this in our revised manuscript.

---

### Official Review · Reviewer_qupx · 2021-07-16

**Rating:** 7
**Confidence:** 2

**Summary:**

This paper proposes a neural framework for manipulating voice, pitch, and speed of speech signals trained in a self-supervised manner without any labels such as text or speaker information. It does so by using features from wav2vec 2.0 and their newly proposed Yingram, with data augmentation on unrelated dimensions to focus on extracting of linguistic information from wav2vec and pitch information from Yingram. The framework can perform zero-shot voice conversion, even to unseen languages, with the use of test-time self-adaptation, which updates the wav2vec feature via backprop using the reference utterance from the unseen language. Results in voice conversion, pitch shifting, and time-scale modification are impressive.

**Ethical Concerns:**

2. Raise safety or security concerns. For example: is there a risk that applications could cause serious accidents or open security vulnerabilities when deployed in real-world environments?

Voice conversion can affect speaker id based security.

7. Deceive people in ways that cause harm. For example: could the approach be used to facilitate deceptive interactions that would cause harms such as theft, fraud, or harassment? Could it be used to impersonate public figures to influence political processes, or as a tool of hate speech or abuse?

Zero-shot voice conversion can be used to impersonate anyone.


The authors acknowledge these issues and hint at the need to have a method to detect such results from their work.

**Limitations And Societal Impact:**

The authors note the potential negative societal impact, and hint at the need to have a method to detect such results from their work.

**Main Review:**

# Originality
I am not familiar with work in this area. Using features from wav2vec for generation seems straightforward, and the idea of perturbing features to avoid the model capturing them is definitely not new. Source-filter models have also existed for some time. But I do think combining everything together in this way is an original approach.


# Quality
The submission is high quality and includes comparisons against various baselines on a variety of objective and subjective metrics. There are just a few points I'm curious about.

- In section 5.2, it is noted that Yingram works better in the case when f0 cannot be accurately estimated. But how does it compare in the common case where f0 estimation is decent?
- The use of the Yingram is to disentangle pitch information and allow pitch shift. I am curious whether the voice conversion works fine with just the wav2vec feature, where the pitch is not perturbed. If voice conversion with just wav2vec works just as well, then it is arguable that the contribution of this paper is closer to "introduce a novel Yingram feature to allow for pitch shift (and also test-time self-adaptation)", which, while it is still a good paper, is a lot less exciting.
- It is not clear to me whether the speaker conditional adversarial loss is necessary. The HiFi-GAN vocoder should be able to produce good samples even from an oversmoothed spectrogram.


# Clarity
The submission is extremely well written. The prose is solid and sections flow well. Enough information is provided to reproduce the work.


# Significance
The results, especially the demos, are impressive. The ability for zero-shot transfer of speaker voice even to unseen languages can have major impact on content dubbing. It would be more exciting if textual information can be controlled as well, as noted in the conclusions.


**Time Spent Reviewing:**

3

---

> ### Author Response · Authors · 2021-08-09
> **Response to Reviewer qupx**
>
> We thank reviewer_qupx for the constructive and kind comments.
> Below are the responses to each of your concerns.
>
>
> \
> In section 5.2, it is noted that Yingram works better in the case when f0 cannot be accurately estimated. But how does it compare in the common case where f0 estimation is decent?
>
> **Authors’ response**: This is absolutely a fair question. The answer is that they sound pretty much identical when f0 estimation is decent. Because the f0 trackers are quite reliable in most of the cases when there is no creaky voice, it is still a good option for representing pitch. The downside of traditional f0 trackers is that they are not trainable, whereas with Yingram, one can train the network to take noisy samples and estimate clean speech samples. By doing so, we can make noise resilient pitch trackers. In our preliminary results, we have found that the network can be trained to track the pitch information from noisy speech samples by simply injecting noise in the source speech in the training stage.
>
> \
> The use of the Yingram is to disentangle pitch information and allow pitch shift. I am curious whether the voice conversion works fine with just the wav2vec feature, where the pitch is not perturbed. If voice conversion with just wav2vec works just as well, then it is arguable that the contribution of this paper is closer to "introduce a novel Yingram feature to allow for pitch shift (and also test-time self-adaptation)", which, while it is still a good paper, is a lot less exciting.
>
> **Authors’ response**: The reason why we shift the pitch is because pitch takes a critical role for perceived speaker similarity. For example, if we convert the voice of a male speaker to a female speaker without matching the pitch of the female speaker, it is highly likely that it wouldn’t sound like the female speaker as it would sound like the female speaker speaking in a very low pitched voice. If we do not use Yingram, the network will have to guess what the pitch is from linguistic information, which seems quite challenging.
>
> \
> It is not clear to me whether the speaker conditional adversarial loss is necessary. The HiFi-GAN vocoder should be able to produce good samples even from an oversmoothed spectrogram.
>
> **Authors’ response**: As a matter of fact, it was reported in HiFi-GAN paper and some TTS papers that HiFi-GAN cannot generate good enough samples from an oversmoothed spectrogram. Therefore, better quality samples can be achieved by adversarially training the network, which makes the output matching the realistic distribution of ground truth mel spectrograms.
>
> \
> Zero-shot voice conversion can be used to impersonate anyone.
> The authors acknowledge these issues and hint at the need to have a method to detect such results from their work.
>
> **Authors’ response**: We agree with the reviewer’s ethical concerns and the potential misusing scenarios. To prevent this, we are studying anti-spoofing algorithms and the accuracy is about 90%. In addition, we’d like to reiterate what we have mentioned to reviewer_pDDj. Voice conversion technology has both sides of a coin. While there can be potential negative misuse cases, there are also positive sides. For example, people who have lost their voices might be able to recover their original voice. In addition, it can be used for those who need privacy preservation.

---

### Official Review · Reviewer_pDDj · 2021-07-31

**Rating:** 6
**Confidence:** 4

**Summary:**

The authors propose a speech autoencoder based on the source-filter model of speech production.

They propose to use pre-trained and fixed feature extraction modules based on wav2vec2 and a newly proposed "Yingram" feature. Using different layers of the w2v2 network, they extract a linguistic embedding and a speaker embedding. An energy feature is computed directly from the input mel spectrogram. The Yingram feature, based on the Yin algorithm, is a pitch feature that encompasses both the fundamental frequency and its harmonics.

The synthesis network is composed of a filter and source generator which are conditioned on subsets of the above features using domain knowledge. The source and filter representation are combined together and trained with a deterministic regression loss on the target mel spectrogram, as well as a GAN-based adversarial loss to compensate for faulty independence assumptions assumptions induced by the diagonal laplace distribution implied by the L1 distance.

A desired end-goal of this autoencoder is to gain control over speaker identity and pitch characteristics in the reconstruction (e.g. a voice conversion or pitch shifting task). Due to redundant information encoded by the wav2vec2 and Yingram features, the authors propose to use data augmentation in feature space to provoke invariance to speaker identity in the wav2vec2 network and invariance to linguistic and speaker properties in the Yingram feature.

The authors transform the input to wav2vec2 with a parametric equalizer, pitch randomization, and formant shifting. This encourages the encodings of the wav2vec2 output to have random fluctuations due to pitch-related phenomena, while the linguistic aspects are kept constant. They similarly transform the Yingram feature with a parametric equalizer followed by formant shifting so that the fundamental frequency is preserved. Since speaker identity is scrambled in both features, they reason the generator will get this information from the speaker embedding.

The system is self-supervised, as it can be trained with no text transcriptions or speaker annotations.

The authors present experiments on downstream tasks:
* zero-shot voice conversion
* reconstruction
* pitch shifting and time-scale modification
and present voice conversion comparisons to baselines VQVC+, AdaIN, and AutoVC.

These analysis features are motivated in contrast to traditional features such as phonetic posteriorgrams (from an ASR network) or an "information bottleneck" latent encoding of the audio from the perspective of:
* data and label efficiency
* reconstruction quality
* language independence

The authors claim that this avoids the trade-off between reconstruction quality and disentanglement that affect prior work.

Finally, the authors present a method for generalizing a pre-trained model to an unseen language.



**Ethical Concerns:**

No ethical concerns.

**Limitations And Societal Impact:**

As stated in my main review, I believe the work is of limited relevance to general machine learning practitioners since it is built on specific domain expertise in speech processing -- the source filter model, for example.

If instead the focus of the paper is on proposing the "information perturbation" idea then I would expect to see examples of the method working in other domains beyond speech (language, images, etc.) and comparison and contrast to similar methods outside of speech.

The main downside of this work in terms of societal impact is that it can enable misuse, abuse and impersonation. Research into countermeasures for generative models of speech is of crucial importance (and an active area of investigation for multiple groups in the community).

**Main Review:**

Originality: This work is the first I've seen using a pre-trained self-supervised audio network for general analysis/synthesis and voice conversion. I don't believe the "information perturbation" method described is novel, but this may be its first application to speech autoencoding to provoke feature invariances.

Quality and clarity: The manuscript is well written and clearly explained. The voice conversion results are very strong relative to their baselines, and it is particularly surprising how poor the baselines perform on intelligibility. However, I'm not familiar with the current voice conversion state of the art, and there are many VC models I know of that were not mentioned.

Significance: The relevance of high quality mel spectrogram analysis and synthesis using language independent methods, based on domain knowledge and inductive biases such as the source-filter model is of high relevance to speech practitioners and researchers. I imagine this work will encourage further exploration of voice conversion tasks, as well as inspiring work in text-to-speech synthesis, bandwidth extension, speech enhancement / separation, and more.

Since the work is built on speech-specific intuition (such as the source-filter model), I expect it to be of relatively low significance to general machine learning practitioners, or those in other disciplines within machine learning. As such, I'm not sure that NeurIPS is the best venue for this work, and think a speech conference would be a better fit.

Missing related work: The related work section is missing many references to current voice conversion methods: ConvS2S-VC, StarGAN-VC, CycleGAN-VC, VoiceGrad, and their variants, for example.

Learning invariances through data augmentation is not a new idea in general, though this may be its first application to speech autoencoding. I think it may be confusing and inaccurate to claim this is a novel technique, so I suggest these claims be tempered. For example, consistency regularization methods like UDA are closely related:

Xie, et al. "Unsupervised Data Augmentation for Consistency Training" (NeurIPS 2020)

Questions and clarifications:
* Please indicate the dimensionality/frequency of the wav2vec2, Yingram and speaker embedding features and compare the dimensionality to other methods that use information bottlenecks.
* Please indicate the number of male/female speakers in the train-360 subset of LibriTTS you use after filtering to greater than 15 minutes.
* Yingram vs. $f_0$: Manually selecting 30 $f_0$ losses to evaluate Yingram against is a biased method of selection and it's not clear what this is meant to demonstrate. Could you explain the motivation behind it? I am left wondering what happens if you use a random sample? Or what happens if you picked the 30 cases where $f_0$ performed the best?
* Reconstruction test: Confusing that Table 1 and Table 2 are not split by seen vs. unseen speaker. Are these tables the combined results of both seen and unseen speakers?
* "Note that in every voice conversion experiment, we shifted the median pitch of a source utterance to the median pitch of a target utterance by shifting the scope of Yingram" -- please explain the motivation of this in the manuscript, as it's not clear why this was done.
* Pitch shift and time-scale modification results: the MOS metrics don't assess whether the desired pitch shift or TSM was applied. How precise was NANSY at controlling these effects?
* I don't believe the impact of the adversarial losses on the performance of the model was mentioned in the text -- it would be very interesting to see an ablation of the adversarial losses.

The test-time self adaptation method was difficult for me to understand because the prose refers to the "input parameters" (which seems like a contradiction to me?). I was unclear whether the feature itself was being optimized or the weights of the wav2vec2 network were being fine-tuned to the target language. Is this an accurate description of how you would use this method to improve voice conversion in an unseen language?
1. For an unseen language, source spectrogram $M_s$, you generate an $\hat{M_s}$ using the network, compute the L1 distance.
2. Compute the gradient of the computed wav2vec2 feature $W_s = \textrm{wav2vec2}(M_s)$ (not the weights of the wav2vec2 network).
3. Optimize the **feature** $W_s$ with gradient descent such that it achieves a better mel spectrogram reconstruction to get $\hat{W_s}$.
4. Take a target speaker spectrogram $M_t$, and compute a speaker embedding for the speaker.
5. Run your generator network again with $\hat{W_s}$, $\textrm{Yingram}(M_s)$ and $\textrm{speaker}(M_t)$ as inputs to get a mel spectrogram in the source speaker's language with the target speaker's voice.

Is that correct?

**Time Spent Reviewing:**

4 hours

---

> ### Author Response · Authors · 2021-08-09
> **Response to Reviewer pDDj (1/2)**
>
> We thank reviewer_pDDj for the extensive and constructive comments. We will try to reflect on the ideas and suggestions to make a better paper. Below are the responses to each of your concerns.
>
>
> \
> Originality: This work is the first I've seen using a pre-trained self-supervised audio network for general analysis/synthesis and voice conversion. I don't believe the "information perturbation" method described is novel, but this may be its first application to speech autoencoding to provoke feature invariances.
>
> **Authors’ response**: Thank you for noting our work as the first work of self-supervised audio network for general analysis/synthesis and voice conversion. It is indeed the key concept of our work.
> The idea of information perturbation is “seemingly” not novel because of its simplicity. However, it is simple and “effective”. We strongly believe the simplicity of the method should be considered novel. In addition, the proper selection of information perturbation functions is one of the keys to the success of our work. We believe it will bring wider attention to other researchers who are working on speech related applications. In addition, by choosing proper information perturbation functions, we believe that the researchers would be able to design an arbitrary generative model with controllable input features and this is not limited only to speech domain.
>
> \
> Significance: The relevance of high quality mel spectrogram analysis and synthesis using language independent methods, based on domain knowledge and inductive biases such as the source-filter model is of high relevance to speech practitioners and researchers. I imagine this work will encourage further exploration of voice conversion tasks, as well as inspiring work in text-to-speech synthesis, bandwidth extension, speech enhancement / separation, and more.
> Since the work is built on speech-specific intuition (such as the source-filter model), I expect it to be of relatively low significance to general machine learning practitioners, or those in other disciplines within machine learning. As such, I'm not sure that NeurIPS is the best venue for this work, and think a speech conference would be a better fit.
>
> **Authors’ response**: We agree with the reviewer’s opinion that the speech-specific intuition such as source-filter decomposition idea used in this paper might not influence the whole audience of NeurIPS. However, we do not consider the source-filter decomposition as the main contribution of this paper. In addition, there have been a significant number of highly influential papers that were published in NeurIPS that are specifically related to speech applications such as [1,2,3,4,5,6,7], and many more.
> Furthermore, we believe the idea of reconstructing signals from self-supervised representations will encourage researchers to try it in their own fields. For example, one could try computer-vision-specific information perturbation functions and encode the residual information with additional encoder (like speaker embedding network) to reconstruct an image from self-supervised ViT features.
> We also believe the test-time self-adaptation method can be adapted by other feature-based generation applications from other domains.
>
> [1] Kumar, Kundan, et al. "Melgan: Generative adversarial networks for conditional waveform synthesis." \
> [2] Kong, Jungil, Jaehyeon Kim, and Jaekyoung Bae. "Hifi-gan: Generative adversarial networks for efficient and high fidelity speech synthesis." \
> [3] Jia, Ye, et al. "Transfer learning from speaker verification to multispeaker text-to-speech synthesis." \
> [4] Défossez, Alexandre, et al. "Sing: Symbol-to-instrument neural generator." \
> [5] Serrà, Joan, Santiago Pascual, and Carlos Segura. "Blow: a single-scale hyperconditioned flow for non-parallel raw-audio voice conversion." \
> [6] Arik, Sercan O., et al. "Neural voice cloning with a few samples." \
> [7] Baevski, Alexei, et al. "wav2vec 2.0: A framework for self-supervised learning of speech representations."
>
> \
> Missing related work: The related work section is missing many references to current voice conversion methods: ConvS2S-VC, StarGAN-VC, CycleGAN-VC, VoiceGrad, and their variants, for example.
>
> **Authors’ response**: Thank you for pointing this out. There are numerous voice conversion papers emerging recently so it was hard to mention all of them in the related work section. We will try to cite some of the CycleGAN-based approaches in our revised manuscript.
>
> \
> Learning invariances through data augmentation is not a new idea in general, though this may be its first application to speech autoencoding. I think it may be confusing and inaccurate to claim this is a novel technique, so I suggest these claims be tempered. For example, consistency regularization methods like UDA are closely related:
> Xie, et al. "Unsupervised Data Augmentation for Consistency Training" (NeurIPS 2020)
>
> **Authors’ response**: Thank you for pointing out the previous work. We were not aware of the mentioned UDA paper. It seems like the paper shares a common idea in that it tries to learn invariance through randomizing inputs. We’ll cite the paper as a relevant approach to our method. However, as far as we are concerned, we have not found an “information perturbation” strategy used for “generation” tasks. In particular, the idea of using the “information perturbation” can be used for unsupervised controllable generation. By choosing proper domain specific “information perturbation” functions, we believe that one can achieve controllable generative modeling in an unsupervised way.
>
> \
> Questions and clarifications: \
> Please indicate the dimensionality/frequency of the wav2vec2, Yingram and speaker embedding features.
>
> **Authors’ response**: The dimensionality of the wav2vec feature is 1024. We modified the wav2vec2 feature extractor (1d-convolution layer parts) to output feature vectors with roughly 22050/256 = 86.13hz. The dimensionality of Yingram is 984. The frequency is the same with wav2vec features. The dimensionality of speaker embedding is 192. It is a single vector.
>
> \
> Compare the dimensionality to other methods that use information bottlenecks.
>
> **Authors’ response**: AUTOVC, for example, reduces channel dimension to 32. The time dimension is downsampled by the factor of 32. In contrast, the channel dimension of our network is 128 and there is no downsampling process.
>
> \
> Please indicate the number of male/female speakers in the train-360 subset of LibriTTS you use after filtering to greater than 15 minutes.
>
> **Authors’ response**: There were 838 (male: 436, female: 402) speakers after filtering.
>
> \
> Yingram vs. f0: Manually selecting 30 f0 losses to evaluate Yingram against is a biased method of selection and it's not clear what this is meant to demonstrate. Could you explain the motivation behind it? I am left wondering what happens if you use a random sample? Or what happens if you picked the 30 cases where f0 performed the best?
>
> **Authors’ response**: The motivation behind this experiment is that we wanted to see if Yingram can actually reconstruct the creaky voice samples in which the f0 tracker fails to estimate f0. We don’t think f0 is a bad feature for synthesizing speech in general. But in the production stage, even a small failure leads to a bad experience for listeners. We have also tried to find samples that have failed to reconstruct speech samples when using Yingram, but we could find little to no samples that failed to faithfully reconstruct the original input.
> If we use random samples for ABX test, it is very likely that there is only a slight difference between using f0 and Yingram because most of the speech samples do not contain creaky voice and the speech samples can still be well reconstructed with f0. What we wanted to make sure of was whether we can reconstruct the samples that were failed by f0 and emphasize the cases when f0 tracker fails for creaky voices.
>
> \
> Reconstruction test: Confusing that Table 1 and Table 2 are not split by seen vs. unseen speaker. Are these tables the combined results of both seen and unseen speakers?
>
> **Authors’ response**: For the English test, it is the combined results of both seen and unseen speakers. There weren’t very many differences between the two settings. The results were as follows: \
> (MOS, DMOS) Seen setting:  (4.17, 1.68), Unseen setting: (4.19, 1.79)
> For the multilingual settings, all speakers were seen speakers during training.
>
> \
> "Note that in every voice conversion experiment, we shifted the median pitch of a source utterance to the median pitch of a target utterance by shifting the scope of Yingram" -- please explain the motivation of this in the manuscript, as it's not clear why this was done.
>
> **Authors’ response**: Thank you for pointing this out. We will edit our manuscript following your suggestion. The reason why we shift the pitch is because pitch takes a critical role for perceived speaker similarity. For example, if we convert the voice of a male speaker to a female speaker without matching the pitch of the female speaker, it is highly likely that it wouldn’t sound like the female speaker instead would sound like the female speaker speaking in a very low pitched voice.

---

> > ### Author Response · Authors · 2021-08-09
> > **Response to Reviewer pDDj (2/2)**
> >
> > Pitch shift and time-scale modification results: the MOS metrics don't assess whether the desired pitch shift or TSM was applied. How precise was NANSY at controlling these effects?
> >
> > **Authors’ response**: Unfortunately we did not measure the objective score in our experiments. The problem of measuring the performance of controlling pitch and time-scale is that there is no ground-truth to compare with (no ground-truth f0 and no ground-truth time-scaled source). But we think it is accurate enough as you can hear it in the samples provided.
> >
> > \
> > I don't believe the impact of the adversarial losses on the performance of the model was mentioned in the text -- it would be very interesting to see an ablation of the adversarial losses.
> >
> > **Authors’ response**: It definitely took a critical role in enhancing the quality of outputs when we compared two of them in our initial experiments. We did not particularly report the results by comparing the two settings (w/ GANs, w/o GANs) as the effect of GANs has been already reported to bring better performance in previous works such as [1,2,3] and many more. It has also been commonly reported in many TTS literatures that the models trained with L1 loss typically outputs over-smoothed mel spectrograms, which results in making unwanted artifacts when using neural vocoders as the vocoders were not trained using the over-smoothed mel-spectrograms. \
> > [1] Lee, Juheon, et al. "Adversarially trained end-to-end korean singing voice synthesis system." arXiv preprint arXiv:1908.01919 (2019). \
> > [2] Donahue, Jeff, et al. "End-to-end adversarial text-to-speech." arXiv preprint arXiv:2006.03575 (2020). \
> > [3] Lee, Sang-Hoon, et al. "Multi-SpectroGAN: High-Diversity and High-Fidelity Spectrogram Generation with Adversarial Style Combination for Speech Synthesis." arXiv preprint arXiv:2012.07267 (2020).
> >
> > \
> > The test-time self adaptation method was difficult for me to understand because the prose refers to the "input parameters" (which seems like a contradiction to me?). I was unclear whether the feature itself was being optimized or the weights of the wav2vec2 network were being fine-tuned to the target language. Is this an accurate description of how you would use this method to improve voice conversion in an unseen language?
> >
> > **Authors’ response**: It is absolutely accurate in every detail except that wav2vec2 network takes a raw waveform, not a mel spectrogram. It is the feature itself that is being optimized.
> >
> >
> > \
> > As stated in my main review, I believe the work is of limited relevance to general machine learning practitioners since it is built on specific domain expertise in speech processing -- the source filter model, for example.
> > If instead the focus of the paper is on proposing the "information perturbation" idea then I would expect to see examples of the method working in other domains beyond speech (language, images, etc.) and comparison and contrast to similar methods outside of speech.
> >
> > **Authors’ response**: We resonate with the reviewer’s concern in a way that it would not be able to influence researchers working in other fields as much as general machine learning papers would do. However, we would like to reiterate that NeurIPS calls for application papers and it is specifically stated in its official webpage (https://nips.cc/Conferences/2021/CallForPapers) as follows: Applications (e.g., speech processing, computational biology, computer vision, NLP).
> > Furthermore, we believe the proposed methods of this work such as “information perturbation” and “test-time adaptation” are not specifically constricted to speech applications. With the proper selection of “information perturbation” functions it can be used in many applications that require controllable generative modeling especially in an self-supervised way.
> >
> > \
> > The main downside of this work in terms of societal impact is that it can enable misuse, abuse and impersonation. Research into countermeasures for generative models of speech is of crucial importance (and an active area of investigation for multiple groups in the community).
> >
> > **Authors’ response**: We are fully aware of the potential negative societal impact this work can bring. To prevent this, we are actually studying anti-spoofing algorithms and the accuracy is about 90%. Voice conversion technology has both sides of a coin. While there can be potential negative misuse cases, there are also positive sides. For example, people who have lost their voices might be able to recover their original voice. In addition, it can be used for those who need privacy preservation.

---

### Decision · Program_Chairs · 2021-09-27

**Decision:**

Accept (Poster)

**Comment:**

UPDATE: The revision has been reviewed and the paper has been accepted.  The authors are encouraged to address other potential harms (e.g., bias, privacy) in their final version.

---

It came to the attention of the program chairs and ethics review chairs late in the review process that this paper concerns voice conversion, a technology that can easily be used in applications that deceive people in ways that cause harm (see https://neurips.cc/public/EthicsGuidelines). This concern was mentioned by some of the reviewers. However, the paper had not been flagged for ethics review and therefore did not go through the formal ethics review process.

This paper was subsequently discussed between the ethics review chairs and program chairs.  In the end, a decision was made to conditionally accept the paper.  Ethics reviewers were brought in to help set conditions.  The list of conditions for acceptance is as follows:

1. Meaningful broader impacts statement. Moving beyond superficial discussion of obvious harms into a more detailed and thorough reflection on ethical issues, especially possible broader impact and potential for misuse.

2. Restricted release of model through some type of licensing or form-restricted access (ie. private repo accessed via request, model code and data use restricted by licenses, etc.)

3. Discussion of possible theoretical and practical mitigation strategies for minimizing harm of such technologies in the future. If this is not possible to discuss, include a clear articulation of the limits of such models in the absence of mitigation approaches. It is not necessary to implement the mitigation strategies discussed though some current work in this area should be highlighted by authors in the main text.

The original meta-review from the AC follows.

---

This paper proposes a neural analysis and synthesis training framework, with a novel training strategy based on information perturbation and the help of wav2vec 2.0 and Yingram. All reviewers had good comments on the novelty and significance of the work. Detailed response also addressed the concerns in the initial review comments. I recommend to accept this paper.